# Identification and validation of a blood-based diagnostic lipidomic signature of pediatric inflammatory bowel disease

Samira Salihovic [1], Niklas Nyström [2], Charlotte Bache-Wiig Mathisen [3], Robert Kruse [4], Christine Olbjørn [5], Svend Andersen [6], Alexandra J. Noble[7,8], Maria Dorn-Rasmussen [9,10], Igor Bazov[1], Gøri Perminow [11], Randi Opheim [3], Trond Espen Detlie[12], Gert Huppertz-Hauss[13], Charlotte R. H. Hedin [14,15], Marie Carlson [16], Lena Öhman [17], Maria K. Magnusson [17], Åsa V. Keita[18], Johan D. Söderholm [18], Mauro D'Amato [19,20,21], Matej Orešič [1,22], Vibeke Wewer [9,10], Jack Satsangi[7,8], Carl Mårten Lindqvist [1], Johan Burisch [10,23], Holm H. Uhlig [7,8,24], Dirk Repsilber[1], Tuulia Hyötyläinen [25,27], Marte Lie Høivik [3,27] & Jonas Halfvarson [26,27] ✉

Improved biomarkers are needed for pediatric inflammatory bowel disease. Here we identify a diagnostic lipidomic signature for pediatric inflammatory bowel disease by analyzing blood samples from a discovery cohort of incident treatment-naïve pediatric patients and validating findings in an independent inception cohort. The lipidomic signature comprising of only lactosyl ceramide (d18:1/16:0) and phosphatidylcholine (18:0p/22:6) improves the diagnostic prediction compared with high-sensitivity C-reactive protein. Adding high-sensitivity C-reactive protein to the signature does not improve its performance. In patients providing a stool sample, the diagnostic performance of the lipidomic signature and fecal calprotectin, a marker of gastrointestinal inflammation, does not substantially differ. Upon investigation in a third pediatric cohort, the findings of increased lactosyl ceramide (d18:1/16:0) and decreased phosphatidylcholine (18:0p/22:6) absolute concentrations are confirmed. Translation of the lipidomic signature into a scalable diagnostic blood test for pediatric inflammatory bowel disease has the potential to support clinical decision making.

Inflammatory bowel disease (IBD), comprising Crohn's disease (CD), ulcerative colitis (UC) and IBD-unclassified (IBD-U), is a complex immune-mediated disease, characterized by chronic gastrointestinal inflammation. IBD is typically diagnosed in young adults, but pediatric-onset IBD is becoming increasingly common, and the incidence is rising[1–3]. Diagnosing IBD in children and adolescents can be challenging as symptoms are unspecific and overlap with many other gastrointestinal diseases[4,5]. These diagnostic obstacles, in combination with the absence of a robust diagnostic test for screening of patients with gastrointestinal symptoms, often translate into a diagnostic delay. In common with several other chronic immune-mediated diseases, a delayed diagnosis is associated with disease complications and surgery[6,7]. Recent reports demonstrate that early treatment and disease control increase the chance of treatment success, avoid complications, and improve overall long-term outcomes[8,9].

---

Several plasma or serum biochemical markers have been investigated as biomarkers in the diagnostic pathway in IBD, i.e., identifying those who should be referred for endoscopy and further investigations. Among the clinically established markers, C-reactive protein (CRP) is the most studied and increased CRP, i.e., >5 mg/L, seems to have the best overall diagnostic performance. However, CRP has poor performance in UC, and even though it often correlates with CD activity, reports have shown that 21% of patients with CD do not mount a CRP response[10,11]. Thus, despite the advantages of CRP over other blood-based biochemical markers, the test is far from ideal, and no specific cut-off for ruling-out pediatric-onset IBD in primary care has been established. Fecal calprotectin is another non-invasive biomarker of IBD that is increasingly used. High levels of fecal calprotectin are found in active IBD, and the test is widely recognized as an indicator of gastrointestinal inflammation, particularly reflecting neutrophil activity. However, fecal tests are poorly accepted by some patients, and a previous meta-analysis of fecal calprotectin identified a lower specificity in children compared with adults[12]. Thus, in order to shorten the diagnostic delay and to identify pediatric patients who should be referred for diagnostic work-up, including gastro-ileocolonoscopy, a reliable and easily obtained biomarker of pediatric IBD is needed.

Lipidomics is the large-scale measurement of a broad range of molecular lipid classes in biological specimens that has emerged as a specialized sub-discipline of metabolomics. Altered lipid metabolism has been suggested to be of particular importance for inflammatory bowel disorders[13–19]. So far, most studies that have assessed lipidomic signatures in IBD have been performed in adult populations. There-

fore, caution should be exercised in their extrapolation to children and adolescents, since the metabolic status in children and adolescents is also intrinsically linked to growth, development, and changing physiology[20]. Furthermore, many of these studies included healthy controls on one side of the diagnostic spectrum and IBD patients receiving treatment on the other. Even though this comparison offers insight into pathophysiological mechanisms involved in the transition from a preclinical to a clinical disease status, this contrast does not accurately reflect a diagnostic scenario where patients with gastrointestinal symptoms seek healthcare and are examined[21]. There is, to our knowledge, currently only one lipidomic study of newly diagnosed treatment-naïve children and adolescents[22]. Although the study indicated lipidomic biomarkers to be of potential relevance in pediatric gastroenterology, few patients were included, and the study lacked external validation. Based upon these considerations, we investigated whether lipidomics could improve the diagnostic prediction of pediatric IBD in two independent cohorts of treatment-naïve patients with newly diagnosed IBD and non-IBD symptomatic controls and further confirmed findings in a third cohort (Fig. 1). We also assessed the potential clinical utility of the validated lipidomic signature compared with clinically established biomarkers.

## Results
### Characteristics of pediatric patients in the discovery, validation, and the confirmation cohort
The discovery cohort comprised 58 children with IBD (CD, $n = 44$, UC, $n = 12$ and IBD-U, $n = 2$) and 36 age-comparable symptomatic controls

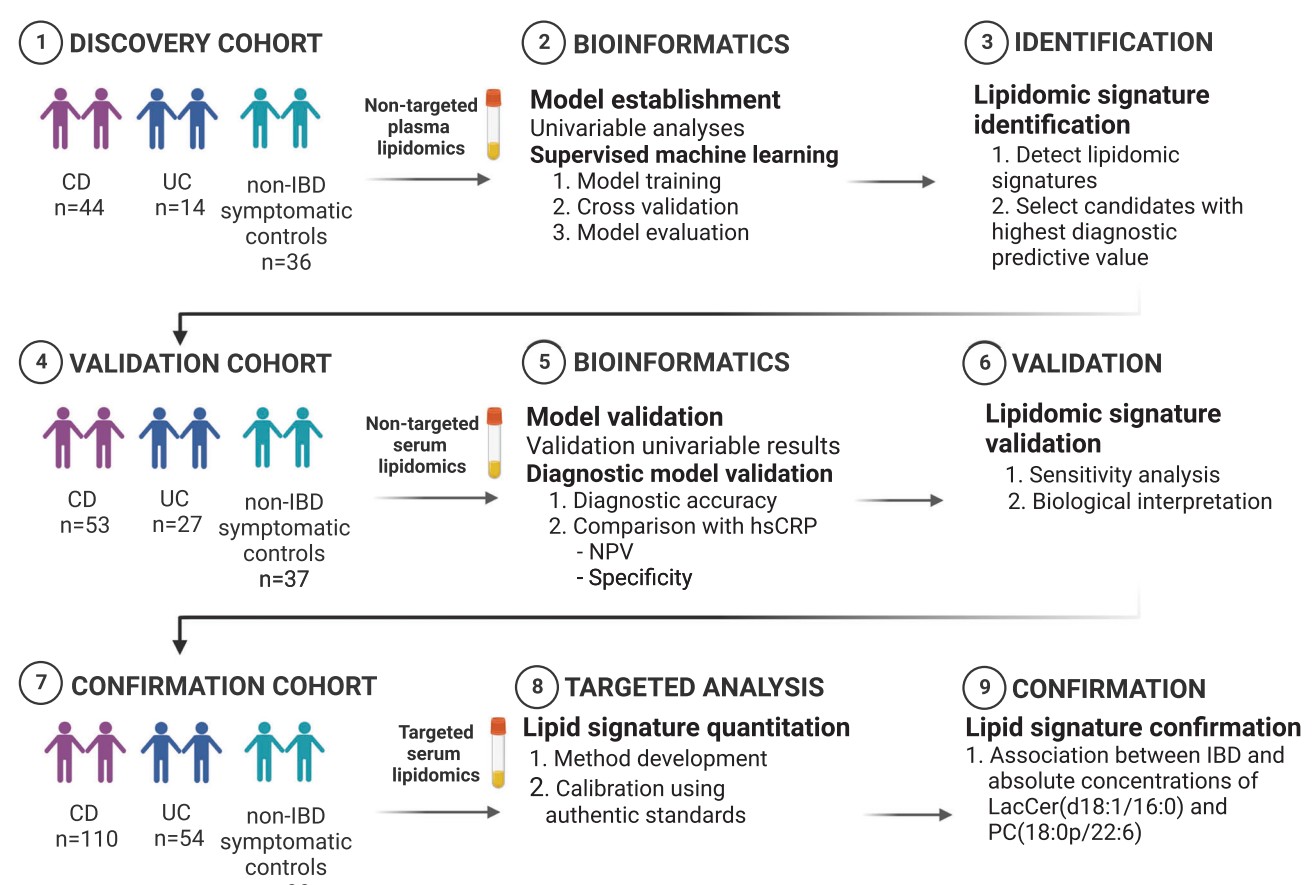

**Fig. 1 | The overall study design.** Illustration of the collection of blood samples from a regional Swedish inception cohort comprising treatment-naïve pediatric patients referred for suspected pediatric IBD. The study findings were validated using the Norwegian population-based IBSEN III pediatric inception cohort and further confirmed in an independent third pediatric cohort. The graphics in this figure were created using Biorender.com. Abbreviations: IBD, inflammatory bowel disease.

**Table 1 | Demographic and clinical characteristics of the pediatric discovery inception cohort and pediatric validation cohort**

| | DISCOVERY | | | VALIDATION | | |
| --- | --- | --- | --- | --- | --- | --- |
| | Uppsala pediatric IBD inception cohort (n = 94) | | | IBSEN III pediatric cohort (n = 117) | | |
| | IBD (n = 58) | Symptomatic controls (n = 36) | *P*-value | IBD (n = 80) | Symptomatic controls (n = 37) | *P*-value |
| Age, median (IQR 1–3) | 15 (12-16) | 12 (9-16) | 0.02 | 14 (10-16) | 12 (7-14) | 0.003 |
| Males, n (%) | 35 (60) | 16 (44) | 0.13 | 45(57) | 23 (62) | 0.63 |
| BMI, median kg/m2 (IQR 1–3) | 19 (18-21) | 17 (16-19) | 0.01 | 18 (16-21) | 17 (16-20) | 0.14 |
| Blood samples, n (%) | 58 (100) | 36 (100) | | 80 (100) | 37 (100) | |
| hsCRP, median (IQR 1–3) | 5 (0.8-12) | 0.5 (0.2–2) | <0.001 | 2.9 (0.6-8.6) | 0.5 (0.3–1.8) | <0.001 |
| Albumin, median (IQR 1–3) | 38 (34-40) | 39 (37-41) | 0.03 | 37 (35-41) | 41 (39-44) | <0.001 |
| Fecal samples, n (%) | 53 (91) | 28 (78) | | 57 (72) | 20 (54) | |
| Fecal Calprotectin, median (IQR 1–3) | 1334 (739-2112) | 69 (40-448) | <0.001 | 861 (260-1801) | 87 (53-124) | <0.001 |
| Subtype IBD (%) | | | | | | |
| Crohn's disease | 44 (76) | NA | | 53 (67) | NA | |
| Ulcerative colitis | 12 (20) | NA | | 21 (25) | NA | |
| IBD-Unclassified | 2 (3) | NA | | 6 (8) | NA | |
| Age at CD diagnosis, n (%) | | | | | | |
| A1a (<10 years) | 6 (13) | NA | | 7 (13) | NA | |
| A1b (10-16 years) | 32 (73) | NA | | 39 (74) | NA | |
| A2 (≥ 17 years) | 6 (13) | NA | | 7 (13) | NA | |
| Location of CD, n (%) | | | | | | |
| L1 (terminal ileum) | 12 (27) | NA | | 8 (15) | NA | |
| L2 (colon) | 17 (39) | NA | | 12 (23) | NA | |
| L3 (ileocolon) | 15 (34) | NA | | 27 (51) | NA | |
| L4 A* (upper GI) | 10 (22) | NA | | 5 (9) | NA | |
| L4 B** (upper GI) | 0 (0) | NA | | 1 (2) | NA | |
| Behavior of CD, n (%) | | | | | | |
| B1 (non-stricturing, non-penetrating) | 41 (93) | NA | | 44 (83) | NA | |
| B2 (stricturing) | 3 (7) | NA | | 8 (15) | NA | |
| B3 (penetrating) | 0 (0) | NA | | 1 (2) | NA | |
| p (perianal disease) | 9 (20) | NA | | 8 (15) | NA | |
| | | | | | NA | |
| PCDAI, median (IQR1-3) | 40 (29-56) | NA | | 15 (10-25) | NA | |
| Age at UC diagnosis, n (%) | | | | | | |
| A1a (<10 years) | 0 (0) | NA | | 5 (25) | NA | |
| A1b (10–16 years) | 12 (86) | NA | | 10 (50) | NA | |
| A2 (≥17 years) | 2 (14) | NA | | 5 (25) | NA | |
| Extent of UC, n (%) | | | | | | |
| E1 (proctitis) | 2 (17) | NA | | 6 (30) | NA | |
| E2 (left sided) | 2 (17) | NA | | 3 (15) | NA | |
| E3 (extensive) | 7 (58) | NA | | 3 (15) | NA | |
| E4 (pancolitis) | 1 (8) | NA | | 8 (40) | NA | |
| PUCAI, median (IQR 1–3) | 43 (30-49) | NA | | 25 (15-40) | NA | |

Statistical analyses were conducted using two-sided Wilcoxon Rank Sum Tests for continuous variables, and chi-squared tests for categorical variables.
*IBD* Inflammatory bowel disease, *CD* Crohn's disease, *UC* ulcerative colitis, *IQR* interquartile range, *PCDAI* Pediatric Crohn's Disease Activity Index, *PUCAI* Pediatric Ulcerative Colitis Activity Index.
*upper gastrointestinal tract proximal of Treitz **upper gastrointestinal tract distal of Treitz.

from the Uppsala pediatric IBD inception cohort (Table 1). The validation cohort included 80 patients with IBD (CD, *n* = 53; UC, *n* = 21 and IBD-U, *n* = 6) and 37 non-IBD symptomatic controls from the population-based pediatric IBSEN III cohort. The confirmation cohort included in total 164 pediatric patients with (CD, *n* = 110; and UC, *n* = 54) and 99 non-IBD symptomatic controls, with 30 of them diagnosed with celiac disease (Table S2).

## Identification of individual molecular lipids that distinguish patients with IBD from symptomatic controls

First, to identify individual molecular lipids that differentiate patients with IBD from symptomatic controls, univariable comparisons were performed. We identified 45 altered molecular lipids in our comparison of IBD with symptomatic controls (Fig. 2a), 65 altered molecular lipids in our comparison of CD with symptomatic controls (Fig. 2b),

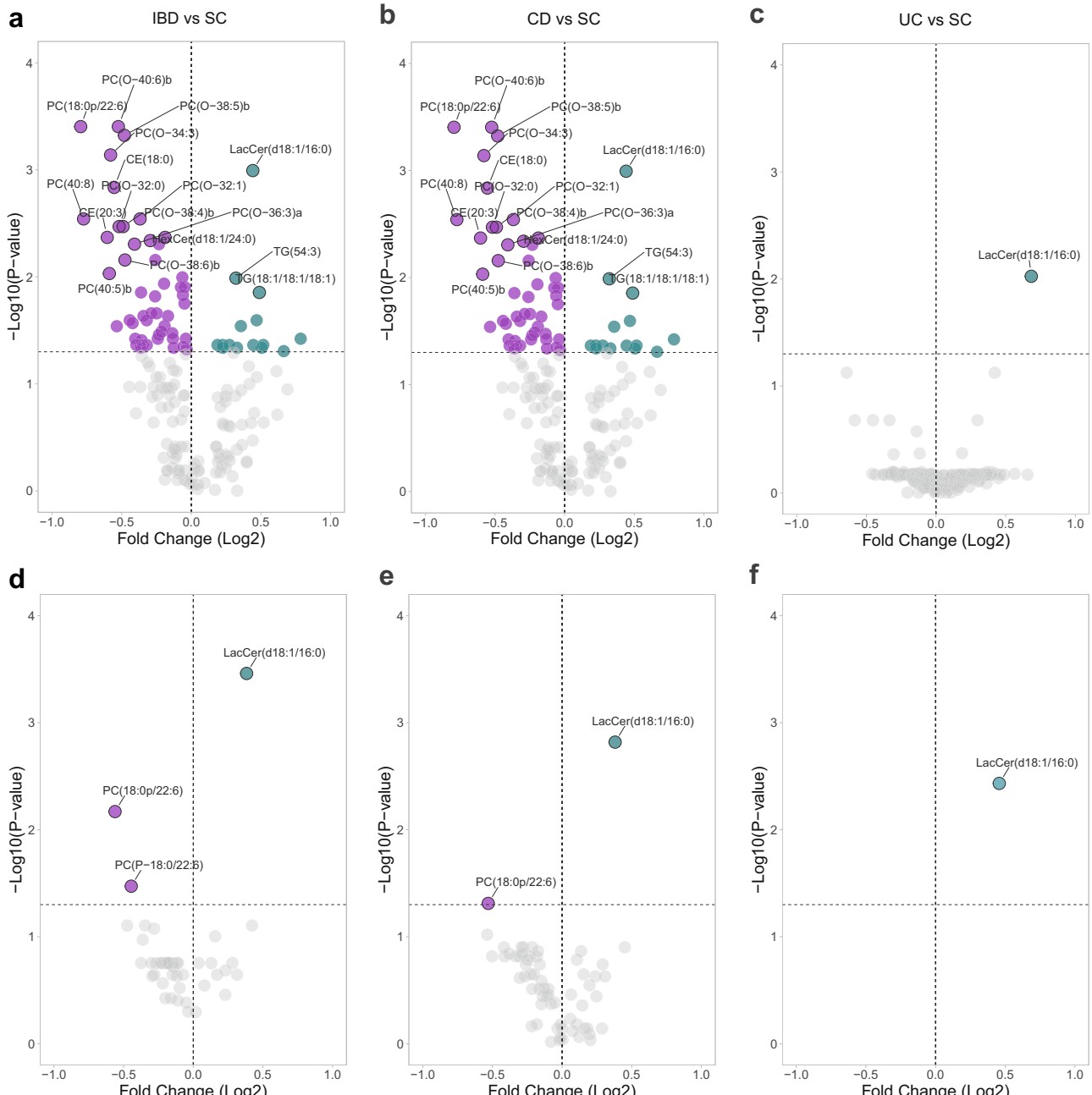

**Fig. 2 | Identification of individual molecular lipids that distinguish patients with IBD from symptomatic controls.** Univariable analysis revealed distinct circulating levels of molecular lipids between IBD, CD, UC and SC in both the discovery cohort (**a**–**c**) and the validation cohort (**d**–**f**), using Wilcoxon rank sum test. The volcano plot represents the Fold Change (Log2) on the x-axis and the corresponding false discovery rate-corrected 2-sided -Log10(*P* value) on the y-axis. IBD inflammatory bowel disease, CD Crohn's disease, UC ulcerative colitis, SC symptomatic controls. Source data are provided as a Source Data file.

and one lipid, LacCer(d18:1/16:0), altered in our comparison of UC with symptomatic controls (Fig. 2c). Molecular lipids were also validated in the IBSEN III cohort using univariable comparisons (Fig. 2d–f). For IBD vs symptomatic controls, three molecular lipids could be replicated in the validation cohort. The corresponding numbers were two for CD and one for UC.

**Performance evaluation of lipidomic signature for IBD diagnosis**
To identify a diagnostic lipidomic signature that differentiates patients with IBD from symptomatic controls, a comprehensive analysis was performed using a set of seven different machine learning algorithms and stacking. The objective was to determine the individual strengths of each algorithm in detecting and classifying lipidomic signature

within the discovery cohort. In general, most of the algorithms were, to a high degree, able to pick-up signatures for classification of patients with IBD vs symptomatic controls in the validation cohort (Table S1). Among the 169 lipids included in these models (Fig. 3a–d), Lac-Cer(d18:1/16:0), PC (18:1p/22:6), TG (56:6), DG (18:1/18:1), PC (35:4), PC (40:6)b, PC(O-32:0), TG (50:5), PC (P-18:0/22:6), TG (48:3), TG (18:1/18:2/18:2)a, PC (38:1), PC (37:3)b, TG (14:0/16:0/18:1), TG (16:0/18:0/18:1), PC (36:5)b, CE (20:5), and TG (46:0) were found to exhibit the highest permuted variable importance across all models.

With the equivalent performance of regularized logistic regression, the SCAD model was used for further optimization. The model for distinguishing patients with IBD from symptomatic controls comprised 30 molecular lipids (Fig. 3a), including 14 PCs, 9 TGs, two LPCs,

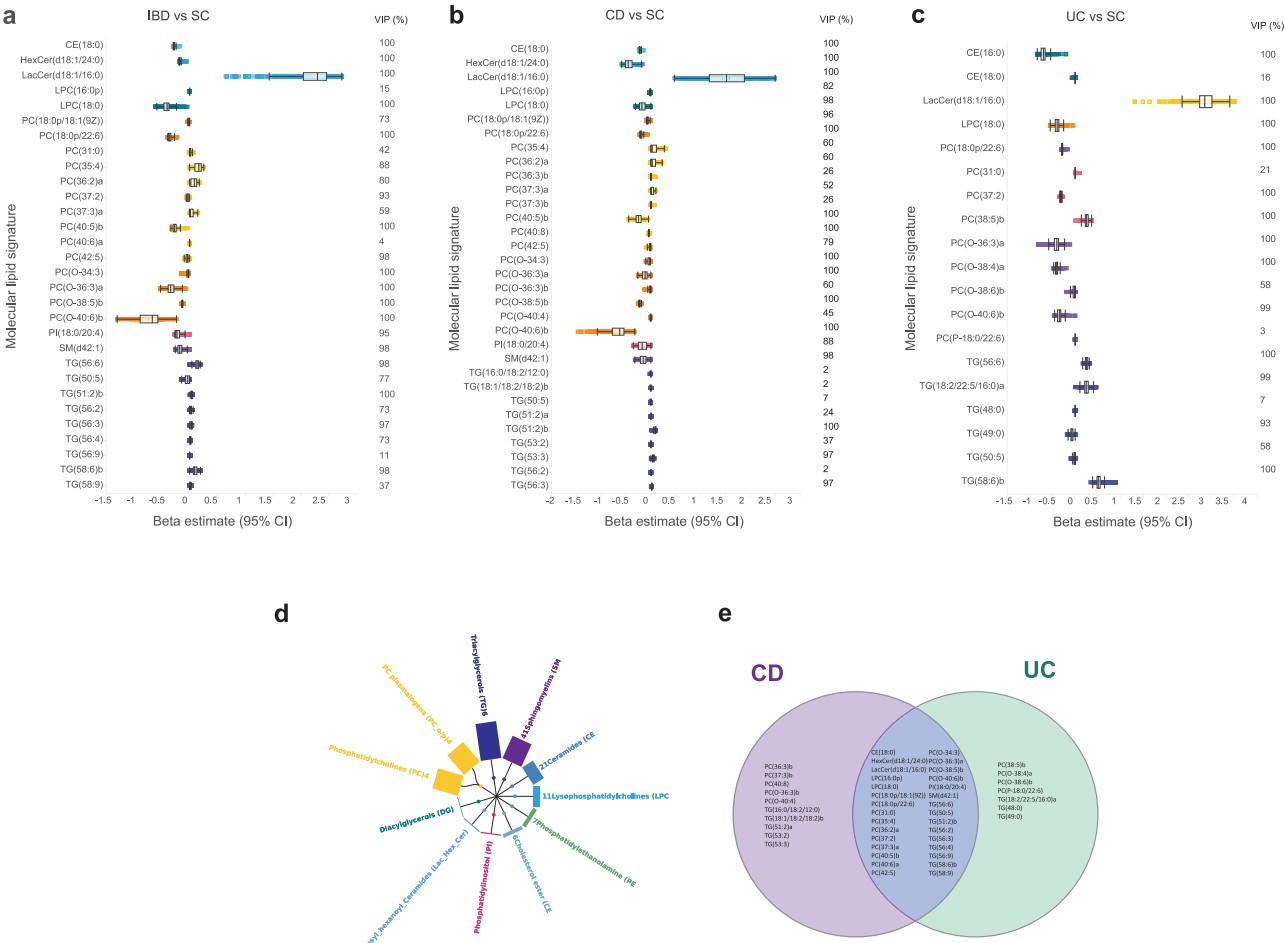

**Fig. 3 | Molecular lipid signatures of IBD.** Variable selection of diagnostic lipidomic signatures using the SCAD model in the discovery cohort (*N* = 94). The bars represent the effect estimates (Beta [95% CI]) of the corresponding molecular lipids selected by the model during fivefold cross-validation. The left and right lines of the boxes indicate the first and third quartiles, the lines in the middle represent the median, and the whiskers extending to the most extreme points within 1.5 times the IQR. Information about the variable importance projection (VIP, %) for each molecular lipid is provided to the right side of each forest plot. **a** In the comparison of IBD vs SC, a diagnostic lipidomics signature consisting of 30 molecular lipids was selected. **b** In the comparison of CD vs SC, a lipidomic signature comprising 32

molecular lipids was selected. **c** In the comparison of UC vs SC, a diagnostic lipidomic signature composed of 19 molecular lipids was selected. **d** Distribution of lipid classes among the 169 lipids that were detected and annotated in both the discovery and validation cohort (*N* = 117). **e** Venn diagram showing the overlap of molecular lipids among the group comparisons: CD vs SC, and UC vs SC. IBD inflammatory bowel disease, CD Crohn's disease, UC ulcerative colitis, SC symptomatic controls, BMI body mass index, hsCRP high-sensitivity C-reactive protein, LacCer(d18:1/16:0) lactosyl ceramide (d18:1/16:0); PC(18:0p/22:6), phosphatidylcholine (18:0p/22:6). Source data are provided as a Source Data file.

CE(18:0), HexCer(d18:1/24:0), LacCer(d18:1/16:0), PI(18:0/20:4), and SM(d42:1) and the effect estimates were bidirectional, with Lac-Cer(d18:1/16:0), several PCs and TGs increased while several PC plasmalogens and SMs decreased. Information about the variable importance projection (VIP) score for each molecular lipid is provided in Fig. 3a–c. The model had an AUC of 0.87 (95% CI 0.79-0.93) in the discovery cohort.

For the diagnostic lipidomic signature of CD vs symptomatic controls, 32 molecular lipids were selected as influential analytes (Fig. 3b) and comprised 16 PCs, 9 TGs, two LPCs, PI(18:0/20:4), SM(d42:1), CE(18:0), HexCer(d18:1/24:0), and LacCer(d18:1/16:0). The model had an AUC of 0.86 (95% CI 0.77-0.93) in the discovery cohort.

For UC vs symptomatic controls, 19 molecular lipids were selected as influential analytes (Fig. 3c) and comprised 9 PCs PC(18:0p/22:6), PC(31:0), PC(37:2), PC(38:5)b, PC(O-36:3)a, PC(O-38:4)a, PC(O-38:6)b, PC(O-40:6)b, PC(P-18:0/22:6), 6 TGs TG(56:6), TG(18:2/22:5/16:0)a, TG(48:0), TG(49:0), TG(50:5), TG(58:6)b. The model had an AUC of 0.90 (95% CI 0.80-0.97) in the discovery cohort. There was a substantial overlap (66%) in lipid signatures between CD and UC (Fig. 3e).

## Validation of a diagnostic lipidomic signature in the IBSEN III cohort

The SCAD model of 30 molecular lipids achieved an AUC of 0.85 (95% CI 0.77–0.92) in discriminating patients with IBD vs symptomatic controls (Table 2). This model had a diagnostic accuracy that was significantly higher than hsCRP (AUC = 0.73, 95% CI 0.63–0.82, *P* < 0.001). We also stratified our validation cohort by subtype of IBD and assessed patients with pediatric CD and UC separately. Compared with the AUC of hsCRP, the diagnostic lipidomic signatures of CD and UC were associated with nominally higher AUC values (0.84, 95% CI 0.74–0.92 vs 0.77, 95% CI 0.67–0.87, *P* = 0.10) and (0.76, 95% CI 0.63–0.88 vs 0.65, 95% CI 0.51–0.78, *P* = 0.06), respectively.

## Establishment of a short and clinically translatable diagnostic model

A signature of many lipids may preclude its translation to clinical practice. Therefore, we went back to the discovery cohort and applied forward stepwise logistic regression to evaluate the potential for establishing a shorter lipidomic signature for discrimination of

**Table 2 | Diagnostic accuracy (area under the curve, AUC) of hsCRP and lipidomic signatures in predicting pediatric inflammatory bowel disease compared to symptomatic controls in the validation cohort**

| Evaluated models | AUC (95% CI) | P value vs hsCRP |
|---|---|---|
| hsCRP | 0.73 (0.63–0.82) | reference |
| Full model (30 molecular lipid species) | 0.85 (0.77–0.92) | 0.001 |
| LacCer(d18:1/16:0) | 0.76 (0.67–0.84) | 0.53 |
| hsCRP + LacCer(d18:1/16:0) | 0.79 (0.70–0.87) | 0.09 |
| PC(18:0p/22:6) | 0.71 (0.61–0.81) | 0.80 |
| hsCRP + PC(18:0p/22:6) | 0.76 (0.66–0.85) | <0.001 |
| PC(18:0p/22:6) + LacCer(d18:1/16:0) | 0.86 (0.78–0.92) | <0.001 |
| hsCRP + LacCer(d18:1/16:0) + PC(18:0p/22:6) | 0.86 (0.77–0.92) | <0.001 |

*AUC* area under the curve, *CI* confidence interval, *hsCRP* high sensitivity C-reactive protein.

patients with IBD from symptomatic controls. We found the highest diagnostic accuracy for a short lipidomic signature, comprising only two molecular lipid species, LacCer(d18:1/16:0) and PC(18:0p/22:6), (AUC = 0.93, 95% CI 0.87–0.98).

The validation confirmed the bidirectional effects of the two lipid species, as evidenced by an increase in LacCer(d18:1/16:0) and a depletion of PC(18:0/22:6) in patients with IBD compared to symptomatic controls (Fig. 4a). Applying this short signature to the validation cohort yielded an AUC of 0.86 (95%CI 0.78–0.92) (Table 2, Fig. 4b, Supplementary Data 1), with a sensitivity of 84% and a specificity of 78% at the optimal cut-off, corresponding to an LR(+) of 3.9 and an LR(−) of 0.20 (Table 3). Compared to the short lipidomic signature, the diagnostic accuracy of hsCRP was significantly lower (AUC 0.73, P < 0.001, Table 2, Fig. 4b, Supplementary Data 1), with a sensitivity of 68% and a specificity of 70%, at the optimal cut-off corresponding to an LR(+) of 2.3 and an LR(−) of 0.5 (Table 3). Also, when comparing the performance of the short lipidomic signature and hsCRP at a fixed sensitivity and specificity, the signature showed higher accuracy (Table 3). The addition of hsCRP to the combination of PC(18:0p/22:6) and LacCer(d18:1/16:0) did not improve the diagnostic accuracy (AUC 0.86) (Table 2, Fig. 4b, Supplementary Data 1). When adding age, sex, BMI, and albumin to the lipid signature, no clinically significant improvement in diagnostic performance was observed (AUC 0.87 vs 0.89) (Fig. 4b, Supplementary Data 1).

To further assess the clinical relevance of the short lipidomic signature, we also evaluated its capacity to reclassify patients with IBD vs symptomatic controls in the validation cohort. The addition of LacCer(d18:1/16:0) and PC(18:0p/22:6) to hsCRP, significantly improved reclassification, as demonstrated by analysis of both NRI and IDI (P < 0.001 for both) (Table 4). Evaluating the net reclassification impact of LacCer(d18:1/16:0) and PC(18:0p/22:6), we observed a substantial improvement of 11% in reclassification of cases with IBD and 14% in reclassification of symptomatic controls, reflecting their dual contribution. This indicates an improved clinical utility of the molecular lipid signature over hsCRP alone. Furthermore, we evaluated the capacity of the molecular lipid signature to rule out IBD in the validation cohort compared with hsCRP. Hereto, the negative predictive value (NPV), i.e., the probability of an individual with a negative test not being diagnosed with pediatric IBD, was compared between hsCRP and the short lipidomic signature at a comparable positive predictive value (PPV). The lipidomic signature yielded a considerably higher NPV (76%) compared to using hsCRP together with its clinically established cut-off (5.0 mg/L) (NPV 40%) at a PPV of 86% for both.

The comparison with fecal calprotectin was limited by the fact that one-third (39/117) of the patients in the IBSEN III cohort did not provide a fecal sample. Among children who provided a stool sample (n = 77), there was no statistically significant difference between the

diagnostic performance of the two lipids (AUC = 0.88 95% CI 0.80–0.95) compared with fecal calprotectin (AUC = 0.93, 95% CI 0.87–0.99, P = 0.22, Table S2, Fig. S1, Supplementary Data 2). Compared to hsCRP, the models utilizing the molecular lipid signature exhibited higher performance in terms of sensitivities, specificities, and likelihood ratios. However, when compared to fecal calprotectin, the molecular lipids showed comparable performance, indicating similar predictive capabilities for pediatric IBD vs symptomatic control (Table S2).

## Sensitivity analysis of short diagnostic signature LacCer(d18:1/16:0) and PC(18:1p/22:6)

To examine the robustness of our findings, sensitivity analyses were conducted. We first analyzed the pair-wise correlations between age, BMI, hsCRP, albumin, fecal calprotectin, LacCer(d18:1/16:0), and PC(18:1p/22:6) and found positive correlations between age and BMI and between LacCer(d18:1/16:0) and hsCRP, albumin, and fecal calprotectin in the discovery cohort (Fig. 4c, Supplementary Data 3). Next, we performed logistic regression to analyze an effect modification for the top two analytes, LacCer(d18:1/16:0) and PC(18:1p/22:6), stratified by age and BMI (Fig. 5a, b, Supplementary Data 4–5). In the case of age, we found a significant interaction between LacCer(d18:1/16:0) and IBD status (P for interaction = 0.002). However, we did not find any significant age interactions between PC (18:1p/22:6) and IBD status (P for interaction = 0.86), indicating that the relationship of LacCer(d18:1/16:0), but not of PC(18:1p/22:6), and IBD is influenced by age. In the case of BMI, we found a significant interaction between LacCer(d18:1/16:0) and IBD status (P for interaction = 0.008), but no significant interaction between PC(18:1p/22:6) and IBD status (P for interaction = 0.55). Taken together, our results indicate that the relationship of LacCer(d18:1/16:0), but not PC(18:1p/22:6), and IBD is influenced by age and BMI.

In order to examine whether the molecular lipids reflect neutrophil activity and gut inflammation per se or are specific to IBD, we assessed the correlation between the molecular lipids and fecal calprotectin levels in the symptomatic controls only. However, no statistically significant correlations were observed between fecal calprotectin and LacCer(d18:1/16:0) (r = 0.28, P = 0.13), or PC(18:1p/22:6) (r = 0.21, P = 0.25).

## Targeted confirmation of the short diagnostic lipid signature in an independent third cohort

We next sought to further confirm the prioritized molecular lipids, LacCer(d18:1/16:0) and PC(18:1p/22:6), in a completely independent third cohort (total n = 263), using targeted quantification by employing calibration curves and surrogate internal standard liquid chromatography coupled to multiple reaction monitoring tandem mass spectrometry. Both molecular lipids were confirmed to be significantly different in our comparison of patients with IBD vs symptomatic controls in the third cohort ($\beta_{LacCer(d18:1/16:0)}$ = 1.08, 95%CI 0.73,1.42, P < 0.001; $\beta_{PC(18:1p/22:6)}$ = −0.55, 95%CI −0.83, −0.27, P < 0.001) (Fig. 6a, b). Also, both molecular lipids exhibited similar magnitudes of effect estimates when stratifying our analyses by subtype of IBD and separately comparing CD ($\beta_{LacCer(d18:1/16:0)}$ = 1.06, 95%CI 0.69,1.43, P < 0.001; $\beta_{PC(18:1p/22:6)}$ = −0.66, 95%CI −0.97, −0.35, P < 0.001), and UC ($\beta_{LacCer(d18:1/16:0)}$ = 1.29, 95%CI 0.78,1.80, P < 0.001; $\beta_{PC(18:1p/22:6)}$ = −0.29, 95%CI −0.64, 0.07, P = 0.12) with symptomatic controls in the third cohort. The comparison of UC vs symptomatic controls was not significant for PC(18:1p/22:6), potentially due to the reduced sample size per subgroup (UC n = 54). To discern whether these molecular lipids serve as markers for inflammatory gastrointestinal diseases generally or are more IBD specific, we compared patients with IBD to the subset of celiac disease patients within the symptomatic controls. We observed significantly increased concentrations of LacCer(d18:1/16:0) ($\beta$ = 1.29, 95%CI 0.78,1.80, P < 0.001) and numerically decreased

**a**

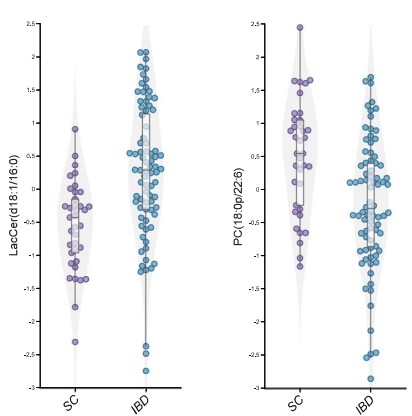

**Fig. 4 | Validation of a diagnostic lipidomic signature in the validation cohort and the relationship with clinical features. a** Plots depicting the log-transformed unit variance scaled distribution of LacCer(d18:1/16:0) and PC(18:1p/22:6) in individuals with IBD compared to symptomatic controls (SC) in the validation cohort (*N* = 117). The upper and lower lines of the boxes indicate the third and first quartiles, the lines in the middle represent the median, and the whiskers extending to the most extreme points within 1.5 times the IQR. Source data are provided as a Source Data file. **b** Receiver operating characteristic (ROC) curve illustrating the diagnostic prediction of pediatric inflammatory bowel disease (IBD) in the validation cohort using logistic regression. The model performance and validity measures were as follows: the area under the curve (AUC) for hsCRP was 0.73 (95% CI 0.63–0.82), while the AUC for the top two validated lipidomic markers, Lac-Cer(d18:1/16:0) and PC(18:0p/22:6), was 0.87 (95% CI 0.80–0.94, *P* = 0.0004). Furthermore, the AUC for hsCRP in combination with the two top validated lipids was 0.87 (95% CI 0.80–0.94). **c** Pair-wise correlations of age, BMI, hsCRP, albumin, fecal calprotectin, LacCer(d18:1/16:0) and PC(18:0p/22:6) among all participants in the discovery cohort were assessed using Pearsons correlation coefficient (**P* < 0.05, ***P* < 0.01 ****P* < 0.001). BMI body mass index, hsCRP high-sensitivity C-reactive protein.

**b**

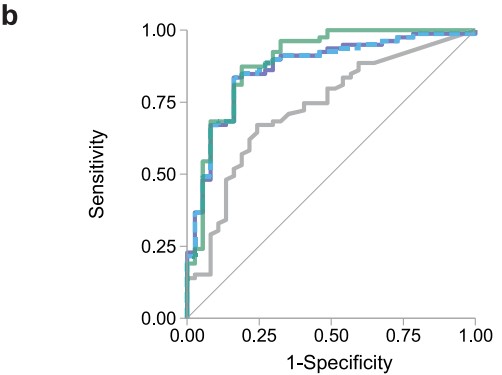

- ■ hsCRP (AUC = 0.73)
- ■ LacCer(d18:1/16:0), PC(18:0p/22:6) (AUC = 0.87)
- ■ hsCRP, LacCer(d18:1/16:0), PC(18:0p/22:6) (AUC = 0.87)
- ■ hsCRP, age, sex, BMI, albumin, LacCer(d18:1/16:0), PC(18:0p/22:6) (AUC = 0.89)
- ― Reference

**c**

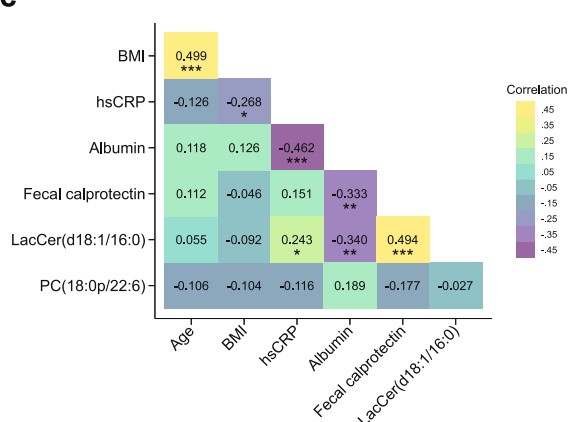

the concentrations of LacCer(d18:1/16:0) did not differ (β = 0.03, 95%CI −0.30, 0.36, *P* = 0.87).

## Discussion

This study undertook a mass spectrometry-based lipidomics analysis of prospectively collected plasma samples from a discovery inception cohort to identify a diagnostic signature of pediatric IBD. By comparing children with IBD and symptomatic children without any discernible evidence of the diagnosis (symptomatic non-IBD controls), we were able to identify a signature of 30 molecular lipids and validate its performance in differentiating pediatric patients with IBD from symptomatic controls in a population-based validation inception cohort. The diagnostic performance of lipidomic signature in the validation cohort was superior to hsCRP (AUC 0.85 vs 0.73). For clinical translation, we demonstrated that a signature of only two molecular lipid species, i.e., LacCer(d18:1/16:0) and PC(18:0p/22:6), was superior to hsCRP and the addition of these molecular lipids to hsCRP, significantly improved the reclassification of patients with IBD from symptomatic controls in the validation cohort. We showed that the lipid signature demonstrates an improved negative predictive value to rule out pediatric IBD when compared to hsCRP at its clinically established cut-off (NPV 76% vs 40%). Among patients providing a fecal sample, the diagnostic performance of the short lipidomic signature and of fecal calprotectin was not materially different. However, a blood-based test may offer enhanced clinical utility, as evidenced by only two-thirds of patients in the validation cohort providing a fecal sample. In an independent third cohort, we confirmed the significant differences in the prioritized molecular lipids (LacCer(d18:1/16:0) and PC(18:1p/22:6)) between patients with IBD and symptomatic controls using a targeted absolute quantification method. Moreover, we demonstrated that these molecular lipids were not broad markers of inflammation but seemed to be more IBD specific. Adopting this short lipidomic signature of only two molecular lipids has the potential to complement existing markers when assessing patients presenting gastrointestinal symptoms suggestive of IBD. This could help clinicians to rule out pediatric IBD and potentially shorten the diagnostic delay.

The observation that circulating LacCer(d18:1/16:0) levels can be used for the diagnostic prediction of IBD is coherent with a previous lipidomics study in pediatric IBD[22]. By analyzing serum samples from 9 children with CD, 10 with UC and 10 children without IBD, with normal fecal calprotectin levels, Daniluk et al. found LacCer(d18:1/16:0) levels to be increased in CD patients. Vila et al. (2023) reported a significant

concentrations of PC(18:1p/22:6) (β = −0.42, 95%CI −0.86, 0.02, *P* = 0.06) in patients with IBD compared to patients with celiac disease. Collectively, these findings confirm the potential of the validated molecular lipids as a set of markers in IBD. In addition, patients with UC, exhibited increased concentrations of PC(18:1p/22:6) (β = −0.43, 95%CI −0.80, −0.06, *P* = 0.02) compared with those with CD, whereas

**Table 3 | Youden index, sensitivity, specificity, positive likelihood ratio (LR), and negative LR of hsCRP compared with two molecular lipids, LacCer(d18:1/16:0) and PC(18:0p/22:6), in predicting pediatric inflammatory bowel disease in the validation cohort**

| Evaluated model | Youden index (J) | Sensitivity (%) | Specificity (%) | LR(+) | LR(-) |
|---|---|---|---|---|---|
| hsCRP | 0.42 | 67.5 | 70.3 | 2.3 | 0.5 |
| PC(18:0p/22:6) and LacCer(d18:1/16:0) | 0.66 | 83.8 | 78.4 | 3.9 | 0.2 |
| hsCRP | NA | 90.0 | 35.1 | 1.4 | 0.3 |
| PC(18:0p/22:6) and LacCer(d18:1/16:0) | NA | 90.0 | 67.6 | 2.8 | 0.1 |
| hsCRP | NA | 29.4 | 90.0 | 3.2 | 0.8 |
| PC(18:0p/22:6) and LacCer(d18:1/16:0) | NA | 66.3 | 90.0 | 7.1 | 0.4 |

The first two rows represent the diagnostic test statistics based on optimal Youden index. Rows three to six show statistics based on fixed sensitivity at 90% and a fixed specificity at 90%.
*hsCRP* high sensitivity C-reactive protein, *LR(+)* likelihood ratio for positive test result, *LR(–)* likelihood ratio for a negative test result.

**Table 4 | Net reclassification improvement and integrated discrimination improvement of pediatric inflammatory bowel disease with hsCRP and with and without two molecular lipids, LacCer(d18:1/16:0) and PC(18:0p/22:6), using the prevalence of IBD (68%) in the validation cohort**

| | No. of patients | | | Reclassified (%) | | |
|---|---|---|---|---|---|---|
| | CRP+lipids classify as SC | CRP + lipids classify as IBD | Total | Downward (%) | Upward (%) | Net reclassified (%) |
| Cases with IBD | | | | | | |
| CRP classifies as SC | 12 | 14 | 26 | 6% | 18% | 11% |
| CRP classifies as IBD | 5 | 49 | 54 | | | |
| Total | 17 | 63 | 80 | | | |
| Symptomatic controls | | | | | | |
| CRP classifies as SC | 25 | 1 | 26 | 16% | 3% | -14% |
| CRP classifies as IBD | 6 | 5 | 11 | | | |
| Total | 31 | 6 | 37 | | | |
| NRI (SE, *P value*) | 0.25 (0.09, *P* = 0.006) | | | | | |
| IDI (SE, *P value*) | 0.24 (0.04, *P* < 0.001) | | | | | |

*hsCRP* high sensitivity C-reactive protein; lactosyl ceramide (d18:1/16:0), LacCer(d18:1/16:0); phosphatidylcholine (18:0p/22:6), PC(18:0p/22:6); *NRI* net reclassification index, *IDI* integrated discrimination index, *SE* standard error.

increase of LacCer (d18:1/16:0) in stool samples of 424 patients with prevalent IBD and 255 non-IBD controls[18]. In the current study, we confirm the observed association between LacCer (d18:1/16:0) and IBD in pediatric populations by examining three cohorts. The fact that two of these cohorts were represented by only treatment-naïve children demonstrates that the increase occurs already at diagnosis. We further extended these findings by showing that the association of Lac-Cer(d18:1/16:0) with IBD was most pronounced in older pediatric patients and in those with a higher BMI. These interactions have not been reported previously and are likely attributed to biological factors linked to childhood growth, development, and changing physiology. The role of sphingolipids in the context of IBD is complex and the mechanisms behind the increased levels of LacCer(d18:1/16:0) remain to be elucidated. Even though we observed increased levels already at diagnosis, it is unclear whether this finding precedes the transition from preclinical IBD to onset of symptoms and an IBD diagnosis. Experimental studies have found various sphingolipids important for plasma membrane stability and for signaling to several receptor molecules[23]. Lactosyl ceramides have, for instance, been found to be significantly enriched in the apical membrane of polarized intestinal epithelial cells[24]. Different pro-inflammatory factors have been shown to activate lactosylceramide synthase to produce lactosyl ceramides, which in turn activate mucosal cell differentiation and maturation[24]. Ceramides can also be transformed into ceramide 1-phosphate, or they can undergo further degradation into sphingosine, which, in turn, can be phosphorylated to produce sphingosine 1-phosphate (S1P). These molecules play a critical role in the regulation of inflammatory processes, and recent drug developments have identified S1P as a

treatment target for IBD, modulating migration of lymphocytes from lymph nodes[25].

We also found that serum PC(18:0p/22:6) levels could be used for diagnostic prediction of IBD. Decreased serum PC(18:0p/22:6) in patients with IBD vs non-IBD symptomatic controls has, to our knowledge, hitherto not been reported as a lipidomic signature in pediatric IBD. In contrast to LacCer(d18:1/16:0), the association of PC(18:0p/22:6) with IBD was not influenced by age or BMI, suggesting that it could be used as an independent predictor of IBD. The pathophysiological role of PCs in IBD is unclear but is likely important since PCs are significantly enriched in the intestine[24,26]. In general, PCs and LPCs make up to 90% of the intestinal mucus and are largely responsible for mucus hydrophobicity[26]. PCs are not only the main structural components of biological membranes but are also involved in cellular signaling. More specifically, the long-chain unsaturated plasmalogen PC(18:0p/22:6) consists of one chain of plasmalogen (alkyl ether) 18:0 at the C-1 position and one chain of omega-3 docosahexaenoic acid 22:6 at the C-2 position. In this study, we detected a depletion of serum PC(18:0p/22:6) levels. Such a depletion of PCs has been observed previously, albeit not for PC(18:0p/22:6) specifically, but for several other PCs and LPCs, including also plasmalogens and other alkyl ether PCs, in plasma and mucosal biopsies from patients with UC[27–30], and in plasma samples from patients with CD[31,32]. Ferru-Clément et al. recently identified several structurally unique lipids (phosphatidylethanolamine ether (O-16:0/20:4), sphingomyelin (d18:1/21:0), cholesterol ester (14:1), very long-chain dicarboxylic acid [28:1(OH)] and sitosterol sulfate) with association to CD when compared to healthy controls, highlighting multiple different biologic pathways including

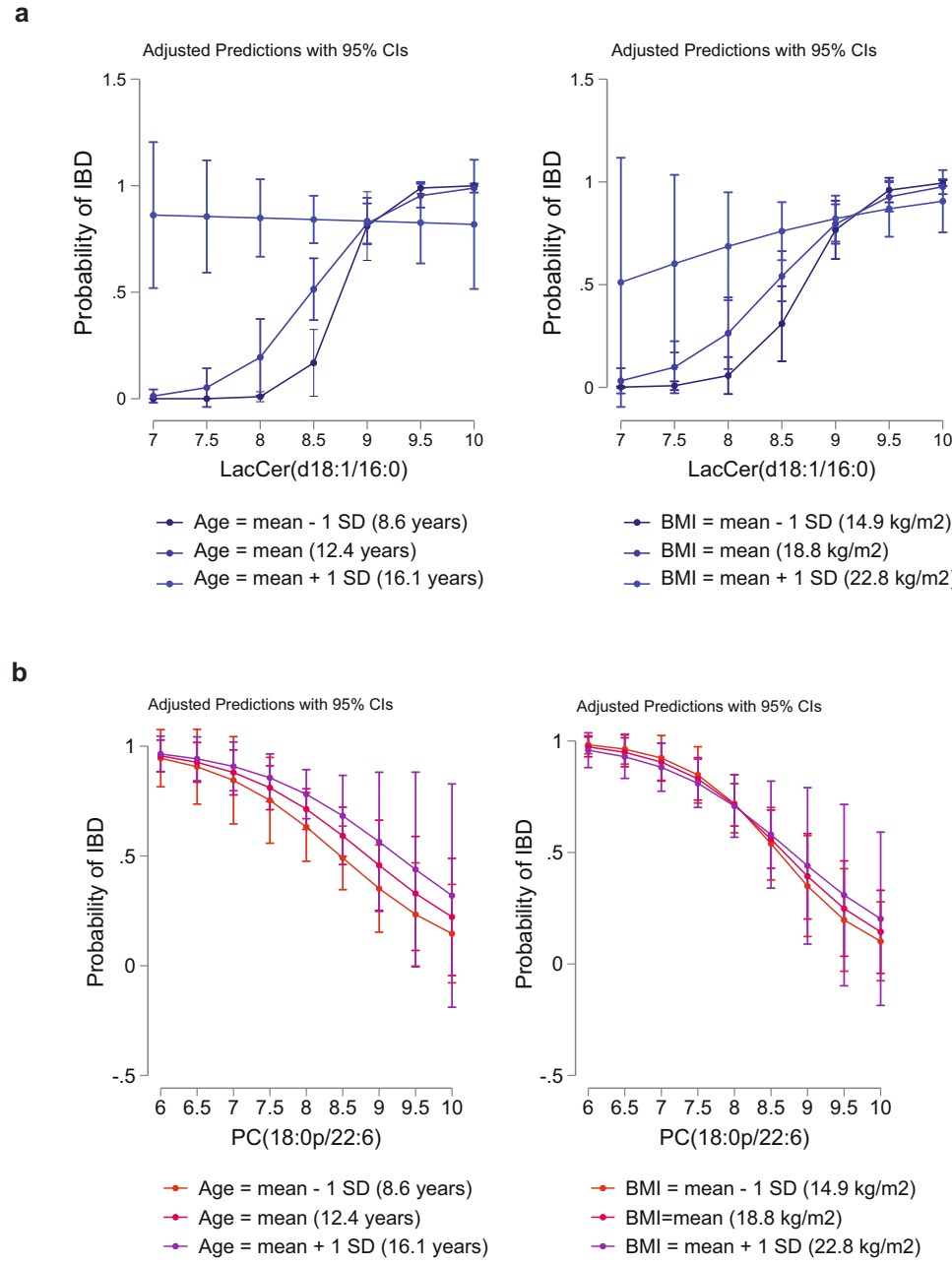

**Fig. 5 | Influence of age and BMI on the association of molecular lipids and IBD. a** Adjusted predictions of IBD at mean and ±1 SD in age and BMI at measured levels of LacCer(d18:1/16:0) in the validation cohort ($N = 117$). **b** Adjusted predictions of IBD at mean and ±1 SD in age and BMI at measured levels of PC(18:0p/22:6) in the validation cohort. BMI body mass index, SD standard deviation.

breakdown of intestinal homeostasis and barrier integrity[19]. Alkyl ether PCs, in addition to their structural roles in cell membranes, are thought to function as endogenous antioxidants, and emerging studies suggest that they are involved in cell differentiation and signaling pathways[33]. These lipids have shown to be endogenous antigens to activate invariant natural killer T cells (iNKT)[34], and associated with autoimmunity[35]. Collectively, our findings of depletion of plasma and serum PC(18:0p/22:6) in pediatric IBD may act as a potential treatment target. This hypothesis is supported by the finding that PC-rich phospholipid supplementation (6 g daily) over three months resulted in an overall decreased inflammatory activity in patients with UC[36].

In a recent metabolomics study using mucosal biopsies and serum samples from a subset of the children in the Uppsala pediatric IBD cohort (treatment-naïve children IBD, $n = 56$, non-IBD controls, $n = 11$) several mucosal metabolites, among them glycerolipids, were found to

differentiate IBD vs non-IBD[37]. In contrast to these observations in mucosal biopsies, no lipid related metabolites were found to be significantly altered in the plasma samples. The discrepancy between the previous findings and the results in this study may be explained by the use of metabolomics vs non-targeted lipidomics and differences in sample size.

This study has several strengths and important limitations. The examination of two independent pediatric inception IBD cohorts, where plasma and serum samples were prospectively collected before the initiation of treatment, is a major strength. The application of both serum and plasma matrices is methodologically advantageous, as it allows for the generalization of results across both matrices. The study is also strengthened by using strict diagnostic criteria to confirm the diagnosis or rule out IBD. The comparison with children with symptoms mimicking IBD demonstrates the clinical relevance of the

**a**

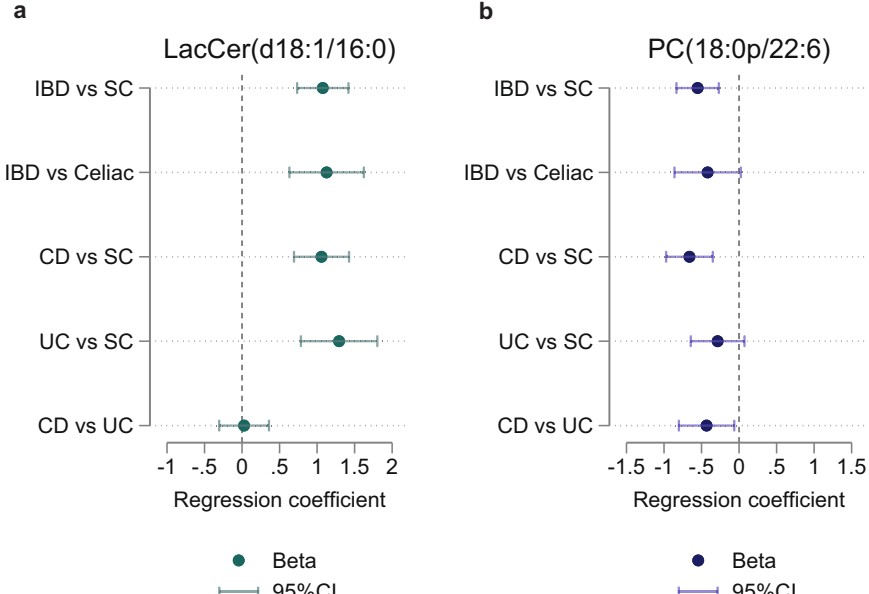

**b**

**Fig. 6 | Targeted analysis of the molecular lipid signature in the confirmation cohort. a** Results from logistic regression analysis of LacCer(d18:1/16:0) among the five different comparisons IBD vs SC, IBD vs Celiac disease, CD vs SC, UC vs SC, and CD vs UC in the confirmation cohort ($N = 263$). Data are presented as beta coefficients with 95% confidence interval lines. All group comparisons, except CD vs UC ($P = 0.87$), were statistically significant ($P < 0.05$). **b** Results from logistic regression analysis of PC(18:1p/22:6) among the five different comparisons IBD vs SC, IBD vs Celiac disease, CD vs SC, UC vs SC, and CD vs UC. Data are presented as beta coefficients with 95% confidence interval lines. All group comparisons, except UC vs SC ($P = 0.12$), were statistically significant ($P < 0.05$). IBD inflammatory bowel disease, CD Crohn's disease, UC ulcerative colitis, SC symptomatic controls, Lac-Cer(d18:1/16:0), Lactosyl Ceramide (d18:1/16:0), PC(18:1p/22:6), phosphatidylcholine (18:1p/22:6). Source data are provided as a Source Data file.

identified circulating lipidomic signature, irrespective of matrix type (plasma vs serum). To identify a diagnostic lipidomic signature and assess its diagnostic accuracy, we used supervised machine learning methodologies and applied the lipidomic signature, i.e., both choice of lipids and the fitted coefficients, as learned from the discovery cohort and applied it to the validation cohort. The fact that we further validated our findings by performing targeted analyses and absolute quantifications of the identified molecular lipids in an independent confirmation cohort increases the generalizability of the findings. Although the number of participants was larger than in many previous studies[22,37], particularly those involving children and adolescents, the number of patients within each Montreal classification category was insufficient to enable meaningful stratified analyses by CD or UC phenotype. Even though there was some overlap between CD and UC associated molecular lipids in the discovery and validation cohorts, our analyses of a larger independent cohort demonstrated that the concentrations of LacCer(d18:1/16:0) and PC(18:1p/22:6) in patients with CD and UC differed from SC. Although we were unable to clearly demonstrate that the molecular lipid signature is unique to IBD, the finding of different concentrations of LacCer(d18:1/16:0) and PC(18:1p/22:6) in patients with IBD compared to patients with celiac disease (another inflammatory disease) indicates that these are not general markers of inflammation. These findings were further supported by the absence of significant correlations between the two molecular lipids and fecal calprotectin levels among symptomatic controls only.

Because of prospective recruitment, we were not able to match patients with IBD and symptomatic non-IBD controls by sex, age, and date of sampling. However, we did find that diagnostic capability of PC(18:1p/22:6) is largely independent of age and BMI. The number of patients in each disease category was based on the inclusion rate, i.e., the number of referred patients during the study period, and was not predefined by a formal estimation of sample size and power analysis. Even though the lipidomic signature outperformed the performance

of hsCRP, fecal calprotectin was associated with a numerically larger AUC among those patients who provided a fecal sample. However, the use of biological material that is difficult to obtain limits its application in health care, as reflected by the observation that one-third of participating children in the validation cohort did not provide a stool sample. To gain further mechanistic understanding, future studies should include patients in remission and evaluate associations of disease activity and retrieve data from follow-up visits of patients in these cohorts and examine the relationship of lipidomic species with therapy response and long-term outcomes, preferably also integrating additional omics data. For clinical translation of the molecular lipid signature, method validation and including standard curve establishment using authentic and isotope-labeled internal and injection standards as well as stability, repeatability, reproducibility, and interlaboratory studies are required for clinical implementation as well as regulatory approval. Furthermore, clinical cut-offs and corresponding likelihood ratios for various clinical scenarios need to be established. Thus, further work is required to ultimately translate our findings into an assay for clinical use.

In conclusion, by examining three independent cohorts, our study identified and validated a lipid signature of two molecular lipid species, LacCer(d18:1/16:0) and PC(18:0p/22:6), that can differentiate between pediatric IBD and children with symptoms indicative of IBD but without any discernible evidence of the diagnosis. Compared with clinically established hsCRP, the lipidomic signature demonstrated increased diagnostic precision and predictive power, and its performance was not materially different from fecal calprotectin. The fact that one-third of patients in the validation cohort did not provide fecal samples indicates that a blood-based test could be associated with improved clinical utility. Taken together, our study suggests a role for Lac-Cer(d18:1/16:0) and PC(18:0p/22:6) in the pathophysiology of IBD and affirms the use of a blood-based lipidomic signature as a tool to be used in combination with existing clinically established markers to rule

out pediatric IBD and guide referral for endoscopy and further investigations.

## Methods

### Study design

We performed a cross-sectional lipidomics study and examined plasma samples from the Uppsala pediatric IBD inception cohort, a prospective single center cohort of children and adolescents referred to the Uppsala University Children's hospital due to clinical suspicion of IBD. By comparing children diagnosed with IBD with symptomatic children who turned out not to have IBD and applying supervised machine learning, we identified a diagnostic lipidomics signature of IBD (Fig. 1). Next, we validated the results from the discovery cohort in serum samples from an independent pediatric population-based inception cohort, i.e., the IBD in South-Eastern Norway – IBSEN III cohort and compared the performance of the lipidomics signature with existing biomarkers, including CRP, albumin and fecal calprotectin. Lastly, we further performed targeted analysis and absolute quantification of the identified molecular lipids with the aim of confirming the findings in an independent third cohort of pediatric Danish, Norwegian and British patients (Fig. 1).

### Study cohorts

**Discovery cohort.** Children and adolescents aged <18 years with suspected IBD who were referred to the Uppsala University Children´s hospital between 2009 and 2018 were consecutively invited to participate. The inclusion criterion was the presence of gastrointestinal symptoms indicative of IBD. Both patients with IBD and symptomatic controls were included at the same point in the diagnostic pathway i.e., before the endoscopic examination. Exclusion criteria included prior IBD diagnosis, systemic infection, ongoing immunosuppressive therapy, previous surgical resection, and treatment with antibiotics within the last three months. After obtaining informed written consent, both blood and fecal samples were collected, and all patients underwent routine diagnostic work-up for IBD in accordance with internationally accepted criteria[11]. The diagnosis of IBD was based on the ESPGHAN/Porto criteria. Children and adolescents who did not meet the diagnostic criteria for IBD were included as non-IBD symptomatic controls. The Paris classification was used to categorize patients according to disease phenotype, the short pediatric Crohn's disease activity index (sPCDAI)[38,39], and pediatric ulcerative colitis activity index (PUCAI)[40,41], to assess clinical disease activity. Correspondingly, we used the simple endoscopic score for Crohn's disease (SES-CD) and the endoscopic Mayo Clinic score to define endoscopic activity.

**Validation cohort.** The validation cohort included children and adolescents from IBSEN III, a population-based inception cohort from a geographically well-defined area, the Norwegian South-Eastern Health Region in Norway. Patients aged <18 years were included from January 2017 until December 2019. Inclusion criteria, clinical work-up, diagnostic criteria, and classification systems were consistent with the discovery cohort and recruited patients were also followed prospectively in accordance with routine clinical procedures. In agreement with the discovery cohort, symptomatic non-IBD controls in the validation cohort were defined as children and adolescents with symptoms of IBD but with no endoscopic or histologic signs of inflammation.

**Confirmation cohort.** In the confirmation cohort, pediatric patients below 18 years, from Denmark[42], Norway and UK were included. The clinical assessment and diagnostic criteria for CD and UC were consistent with those utilized in the discovery and validation cohort. Patients from Denmark and Norway were recruited at diagnosis and subsequently followed prospectively as per routine clinical care. Similar to the discovery and validation cohorts, symptomatic non-IBD

controls from these two countries comprised patients with symptoms of IBD but lacking any endoscopic or histologic evidence of IBD. In the UK cohort, children and adolescents with a previous diagnosis of IBD were included as cases and the pediatric patients with other gastrointestinal disease, i.e., primarily functional gastrointestinal disorders and celiac disease, were included as symptomatic controls.

### Ethical permission

Written informed consent was obtained from each participant or from the participant's parents or legal guardians in the discovery, validation, and confirmation cohorts, and the study was conducted according to the Declaration of Helsinki. Ethical permission was granted by the Uppsala University Ethics Committee, Sweden (2008/395), the South Eastern regional Ethical board, Norway (2015/946), the Ethics Committee of the Capital Region of Denmark (H-20065831), the Danish Data Protection Agency (P-2020-1065), and the Oxford Research Ethics Committee, Reference: 11/YH/0020 and 16/YH/0247.

### Lipidomics

**Sample preparation and analysis.** Plasma samples from the discovery inception cohort and serum samples from the validation cohort (10 μl) were randomized and extracted with a modified version of the previously published Folch procedure[22]. There were no samples with insufficient volume for subsequent lipidomic analysis. In short, 10 μl of 0.9% NaCl and, 120 μl of CHCl3: MeOH (2:1, v/v) containing the internal standards (c = 2.5 μg/mL) was added to each sample. The internal standard solution contained the following compounds: phosphatidylethanolamine (PE(17:0/17:0)), sphingomyelin (SM(d18:1/17:0)), ceramide (Cer(d18:1/17:0)), phosphatidylcholine (PC(17:0/17:0)), lysophosphatidylcholine (LPC(17:0)) and lysophosphatidylcholine (PC(16:0/d31/18:1)), were purchased from Avanti Polar Lipids, Inc. (Alabaster, AL, USA) and triheptadecanoylglycerol (TG(17:0/17:0/17:0)), and cholesteryl ester (CE17:0) were purchased from Larodan AB (Solna, Sweden). The calibration curve solutions contained the following compounds: LPC (18:0), cholesteryl ester (18:1, 9Z), Cer(d18:1/24:0), Cer(d18:0/18:1, 9Z), triglyceride (16:0/16:0/16:0), PC(16:0/16:0), Triglyceride (18:0/18:0/18:0), CE(18:0), LPC(18:1), LPE(18:1), (PC(16:0/18:1), Cer(d18:1)/18:1, 9Z), PC(18:0/18:0), PE(16:0/18:1), CE(18:2, 9Z, 12Z), CE(16:0), DG(18:1). The samples were vortex mixed and incubated on ice for 30 min, after which they were centrifuged (9400 × g, 3 min). 60 μl from the lower layer of each sample was then transferred to a glass vial with an insert, and 60 μl of CHCl3: MeOH (2:1, v/v) was added to each sample. Instrumental analyses were carried out on an ultra-high-performance liquid chromatography quadrupole time-of-flight mass spectrometry method (UHPLC-Q-TOF-MS) from Agilent Technologies (Santa Clara, CA, USA). The analysis was done on an ACQUITY UPLC® BEH C18 column (2.1 mm × 100 mm, particle size 1.7 μm) by Waters (Milford, USA). Mobile phases were as follows A: 10 mM NH4Ac and 0.1% formic acid in water and B: 10 mM NH4Ac and 0.1% formic acid in acetonitrile/isopropanol (1:1). Dual jet stream electrospray (dual ESI) ion source was used and with the ion polarity set to positive mode. An internal standard mixture was used for normalization and lipid-class specific calibration (0.1–10 μg/ml) was used for quantitation as previously described.

**Data pre-processing.** MZmine 2.53 was used for pre-processing of non-targeted lipidomics raw data[43]. First, we performed peak detection with a noise level of 1000 followed by ADAP chromatogram builder with the group intensity threshold at noise level 200, minimum highest intensity 1000, and m/z tolerance 0.009 m/z or 8 ppm. Next, we performed chromatogram deconvolution with local minimum search algorithm with a 70% chromatographic threshold, 0.05 min of minimum retention time range, 5% minimum relative height, 2250 minimum absolute height, 1 as minimum ratio of peak top/edge and peak duration range in minutes from 0.08 to 5.00. We performed

isotopic peak grouping with a m/z tolerance of 0.05 m/z or 5 ppm, tR tolerance was set on 0.05 min, and a maximum charge of 2. Alignment of peak lists, a "Join alignment" algorithm was used with a m/z tolerance as 0.006 or 10.0 ppm and a weight of m/z as 2 with a tR tolerance of 0.1 and a weight of tR 1. Filtering with "Feature list" rows filter was done next in three steps. In the first step, rows that matched all criteria were kept with a retention time of 2–12 and m/z between 369 and 1200. In the second step, rows that matched with all criteria with a tR range of 2–4 and m/z of 800–1200 were removed. In the third step rows that match with criteria with a tR range of 4–8 and m/z of 370–500 were excluded. Gap filling with "Peak finder" was performed with a m/z tolerance of 0.006 m/z or 10.0 ppm, tR tolerance of 0.1 min and with an intensity tolerance of 50%. Although the serum and plasma samples were processed using an identical methodology, prior studies that have examined the variations in lipidomics profiles between plasma and serum derived from the same individuals have reported that serum matrices, which lack clotting activation, show greater variability in the levels of certain lipids compared to plasma matrices[44]. Additionally, the studies revealed that serum shows approximately 20% higher concentrations of specific lipids compared to plasma. Lastly, comparisons with a custom database were made to identify the peak list with compound names. In total, 169 annotated individual lipidomics species were acquired in serum and plasma from the discovery and validation cohorts representing a broad range of lipid classes, including diacylglycerols (DGs), triacylglycerols (TGs), phosphatidylcholines (PCs), phosphatidylethanolamines (PEs), phosphatidylserines (PSs), phosphatidylinositols (PIs), ceramides (Cer), lactosyl or hexosylceramides (Lac Hex Cer), and sphingomyelins (SMs).

**Targeted confirmation of molecular lipid signature using UHPLC-MS/MS.** Plasma and serum samples from the third cohort (10 μl) were randomized and extracted using the same procedure as for the non-targeted analysis as described above. An eight-point calibration curve (range 0.5–1600 ng/mL) was constructed using authentic standards of the two top prioritized molecular lipids LacCer(d18:1/16:0) and PC(18:1p/22:6) purchased from Avanti Polar Lipids, Inc. (Alabaster, AL, USA). An isotopically-labeled form was not commercially available at this time, and therefore we used two chemically similar internal standards, Cer(d18:1/17:0) and PC(17:0/17:0), both eluting closely to the target analytes. Targeted analysis was performed using ACQUITY PREMIER UHPLC I-Class equipped with a ACQUITY PREMIER C18 column (2.1 mm × 50 mm, particle size 1.7 μm) with Xevo TQ-XS Mass Spectrometer operated in positive electrospray ionization (ESI) mode (Waters, Milford, USA). The analysis of the two top molecular lipids was tuned, optimized and finally included multiple reaction monitoring of 3–4 of the most abundant precursor and their respective product ions for quantification and qualification (LacCer(d18:1/16:0) m/z 862.70 > 264.20, 862.70 > 282.20, 862.70 > 520.50, PC(18:0p/22:6) m/z 818.60 > 86.00, 818.60 > 184.00, 818.60 > 508.50, 818.60 > 550.50). The linearity (R2) and relative response factor standard deviation (RRF, %) was for LacCer(d18:1/16:0) 0.985 and 13%, and for PC(18:0p/22:6) 0.997 and 10%.

**C-reactive protein**
In the discovery cohort, hsCRP was measured in clinical routine, and results were available for most (90%) patients. In the validation cohort, hsCRP was assayed in a single batch for all patients at the end of the recruitment period, using a particle-enhanced immunoturbidimetric hsCRP assay (Cardiac C-Reactive Protein (Latex) High Sensitive, Roche Diagnostics) on a Roche Cobas c501 at Uppsala BioLab, Uppsala Clinical Research Center, Uppsala, Sweden.

**Fecal calprotectin**
Fecal samples from patients in the discovery cohort were analyzed for calprotectin as part of clinical routine, using fCAL ELISA Calprotectin

assay (Bühlmann Laboratories AG, Schönenbuch, Switzerland) or LIAISON Calprotectin Assay (Diasorin S.p.A Saluggia, Italy). All fecal samples from the validation cohort were extracted in one batch after all patients had been included, and concentrations were measured using the fCAL ELISA Calprotectin assay (Bühlmann Laboratories AG, Schönenbuch, Switzerland) at Unger Vetlesen Institute, Loivsenberg Diaconal Hospital, Oslo, Norway.

## Statistical analysis
**Data pre-processing.** The lipidomic measurements yielded 663 plasma molecular lipids in the discovery cohort and 687 serum molecular lipids in the validation cohort, of which 169 were matched to known annotations the according to the LipidMaps nomenclature, (www.lipidmaps.org), passed quality control, and were found above the limit of detection (LOD) in both the discovery and validation cohort in >50% of the samples and were uniformly distributed among both the cases and controls. These 169 plasma and serum lipids that were detected in both cohorts were then retained for subsequent statistical analysis. Among the retained lipids, non-detected values were imputed by the lowest limit of detection (LOD/2). The data were log2-transformed and batch-corrected using ComBat[45].

**Analysis of demographic data and clinical cohort characteristics.** Categorical and continuous variables, representing demographic or clinical characteristics, are presented as proportions and median and interquartile range (IQR). Levels of hsCRP, albumin, and fecal calprotectin were log2-transformed before analysis.

**Univariable statistical analyses of individual molecular lipid species.** Univariable analyses of IBD overall and by subtype of IBD, i.e., CD and UC separately, were performed by the Wilcoxon rank-sum test. The number of patients with IBD-U was too small ($n = 2$ in the discovery cohort, $n = 6$ in the validation cohort and $n = 0$ in the confirmation cohort) to allow any meaningful specific analyses and IBD-U was, therefore, merged with UC. $P$ values were adjusted for multiple comparisons using a false discovery rate (FDR) approach[46]. Individual molecular lipids were regarded as being significantly different in the discovery cohort if they showed a $P_{FDR} < 0.05$. Furthermore, those individual molecular lipids were considered validated if they showed a $P_{FDR} < 0.05$ in the validation cohort.

**Multivariable statistical analyses of individual molecular lipid species.** Seven different machine learning algorithms and a stacked model approach was initially employed on all 169 lipids to discern algorithm strengths with regards to finding diagnostic lipidomic signatures in the discovery cohort for evaluation in the validation cohort (for modeling details see Table S1). Model performance was assessed by comparing the area under the receiver operating curve (AUC) averaged over nested cross-validations. Among the various machine learning techniques employed, regularized logistic regression was selected for further optimization since its performance was equivalent to the less interpretability transparent algorithms. Thus, regularized logistic regression was chosen as the algorithm for further signature optimization. For this purpose, regularized logistic regression with Smoothly Clipped Absolute Deviation (SCAD) [*ncvreg* R package][47] was used to limit the number of lipids to more distinct specific signatures. The SCAD models were employed using alpha = 0.1 and a lambda obtained by optimization in internal 5-fold cross-validations. The selection of the predictive lipidomic signature was based on 500 model fits, and all individual molecular lipids with non-zero coefficients in any model are reported. The diagnostic lipidomic signature models were built using data exclusively from the discovery cohort and validated using data from the validation cohort. The top validated differential lipidomic signatures were used for prediction, and AUC with 95% confidence interval (95% CI) was reported. AUCs were

compared using a bootstrap approach (*pROC* package)[48]. The Youden index was used to derive the optimal cut-off value, which was further used to determine sensitivity, specificity, and likelihood ratio for a positive result [LR(+)], and likelihood ratio for a negative result [LR(−)]. Reclassification was assessed using net reclassification index (NRI) and integrated discrimination improvement (IDI)[49]. Positive predictive values (PPV) and negative predictive values (NPV) were also calculated and reported. For sensitivity analysis, pair-wise associations among age, body mass index (BMI), hsCRP, albumin, fecal calprotectin, and molecular lipids were assessed using Pearson correlation coefficients. The moderating effects of age and BMI on the association between molecular lipids and IBD were assessed by introducing interaction terms. Confirmation of results in the third cohort was examined using logistic regression. Statistical analyses were performed using the statistical computing language STATA16 and R (version 4.1.2).

### Reporting summary

Further information on research design is available in the Nature Portfolio Reporting Summary linked to this article.

## Data availability

Lipidomics data are provided as a Source Data file for each cohort. The clinical data associated with the lipidomics dataset are not publicly available because of patient confidentiality. However, the data can be made available for IBD research upon request through a minimal access procedure. This procedure consists of sending a request per email to the corresponding author (Jonas Halfvarson, jonas.halfvarson@regionorebrolan.se) including a copy of the ethics approval. A response will be provided within two weeks. This procedure is installed to ensure that the clinical data are being requested for scientific purposes only and thus complies with the informed consent signed by the participants, since the collected data cannot be used by commercial parties. Source data are provided with this paper.

## Code availability

Statistical analyses were performed in STATA16 (https://www.stata.com). For the exploratory machine learning algorithms R (version 4.1.2) and the following R packages were used: glmnet, ranger, xgboost, nnet, klaR, bonsai, discrim, and stacks, bundled within the parsnip R package of the Tidymodels meta-package described at: https://cran.r-project.org/web/packages/parsnip/index.html. The machine learning algorithms original R packages are described at: https://cran.r-project.org/web/packages/glmnet/index.html, https://cran.r-project.org/web/packages/ranger/index.html, https://cran.r-project.org/web/packages/xgboost/index.html, https://cran.r-project.org/web/packages/nnet/index.html, https://cran.r-project.org/web/packages/klaR/index.html, https://cran.r-project.org/web/packages/bonsai/index.html, https://cran.r-project.org/web/packages/discrim/index.html, and https://cran.r-project.org/web/packages/stacks/index.html, respectively. The final diagnostic models were built using custom code in R (version 4.1.2). The custom code used for this study can be found at the following link: https://github.com/dirkrepsilber/diagnostic_lipidomics_pediatric-ibd (https://doi.org/10.5281/zenodo.10798066).

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

## Acknowledgements

This work was supported by the Swedish Foundation for Strategic Research [RB13-0160 to J.H.], the Swedish Research Council [2020-02021 to J.H.], the Örebro University Hospital research foundation [OLL-890291 to J.H.], NordForsk [90569 to J.H.]. The funders had no role in considering the study design or in the collection, analysis, interpretation of data, writing of the manuscript, or decision to submit it for publication.

## Author contributions

Conceptualization, N.N., D.R., T.H., M.L.H., and J.H.; Methodology, D.R., T.H., S.S., M.L.H., and J.H.; Formal Analysis, S.S., I.B., R.K., D.R., and J.H.; Investigation, S.S., N.N., C.B.W.M., C.O., S.A., A.J.N., M.D-R., G.P., R.O., T.E.D., G.H.H., I.B., R.K., C.M.L., C.R.H.H., M.C., L.Ö., M.K.M., Å.V.K., J.D.S., M.DA., M.O., V.W., J.S., J.B., H.H.U., T.H., M.L.H., and J.H.; Resources, S.S., N.N., C.B.W.M., S.A., C.O., G.P., R.O., T.E.D., G.H.H., T.H., M.L.H., and J.H.; Data Curation, S.S., N.N., C.B.W.M., I.B., R.K., C.M.L., D.R., T.H., M.L.H., and J.H.; Writing – Original Draft, S.S. and J.H.; Writing – Review & Editing, S.S., N.N., C.B.W.M., C.O., S.A., A.J.N., M.D-R., G.P., R.O., T.E.D., G.H.H., I.B., R.K., C.M.L., C.R.H.H., M.C., L.Ö., M.K.M., Å.V.K., J.D.S., M.DA., M.O., V.W., J.S., J.B., H.H.U., T.H., M.L.H., and J.H.; Visualization, S.S, I.B., D.R., and J.H.; Supervision, D.R., T.H., M.L.H., and J.H.; Project Administration, J.H.; Funding Acquisition, M.L.H., and J.H.

## Funding

## Competing interests

Dr Salihovic has no conflicts of interest to disclose. Dr Nyström has served as speaker and/or advisory board member for Abigo, Baxter, Ferring, Fresenius-Kabi, Mylan/Meda, Nutricia, Shire, Takeda, Thermo Fisher Scientific, Tillotts Pharma, and Viatris. Dr Bache-Wiig Mathisen has served as advisory board member for Tillotts Pharma. Dr Andersen has no conflicts of interest to disclose. Dr Olbjørn has no conflicts of interest to disclose. Dr Perminow has served as a speaker and/or advisory borad member for AbbVie. She has also received grant support from Ferring, Tillotts Pharma and Takeda. Dr Opheim has no conflicts of interest to disclose. Dr. Detlie has served as a speaker, consultant, or advisory board member for AbbVie, Ferring, Pfizer, Pharmacosmos, Tillotts, and Vifor Pharma. He has received unrestricted research grants from AbbVie, and Pharmacosmos. Dr Huppertz-Hauss has no conflicts of interest to disclose. Dr Bazov has no conflicts of interest to disclose. Dr Kruse has no conflicts of interest to disclose. Dr Lindqvist has no conflicts of interest to disclose. Dr. C. R. H. Hedin has received speaker fees from Takeda, Ferring, AbbVie, and Janssen, and consultancy fees from Pfizer. She has acted as local principal investigator for clinical trials for Janssen and GlaxoSmithKline. She is PI on projects at the Karolinska Institutet partly funded by investigator-initiated grants from Takeda and Tillotts. None of these activities have any relation to the present study. Dr Carlson has received speaker's fees from ViforPharma. She is the national PI for clinical trials for AstraZeneca. None of these activities have any relation to the present study. Dr Öhman has received financial support for research from Genetic Analysis A.S., Biocodex, Danone Research and AstraZeneca and served as Consultant/Advisory Board member for

Genetic Analysis A.S., and as a speaker for Biocodex, Janssen, Ferring Pharmaceuticals, Takeda, AbbVie, Novartis, Avanos, and MEDA. Dr Magnusson has no conflicts of interest to disclose. Dr Keita has no conflicts of interest to disclose. Dr Söderholm has no conflicts of interest to disclose. Dr D'Amato has received unrestricted research grants and serves as consultant for QOL Medical. Dr Orešič has no conflicts of interest to disclose. Dr Noble has no conflicts of interest to disclose. Dr Satsangi has consultancy fees from Janssen. Current research support from The Helmsley Trust, CCUK, and EC Horizon 2020 programme. Dr Uhlig has received research support or consultancy fees from Janssen, UCB Pharma, Eli Lilly, Boehringer Ingelheim, Pfizer, AbbVie, BMS Celgene, GSK, OMass and MiroBio. Dr Dorn-Rasmussen has no conflicts of interest to disclose. Dr Wewer has no conflicts of interest to disclose. Dr Burisch reports personal fees from AbbVie, Celgene, Pfizer, Samsung Bioepis, Pharmacosmos, Ferring, and Galapagos; grants and personal fees from Janssen, MSD, Takeda, Tillots Pharma, and Bristol Myers Squibb; and grants from Novo Nordisk. Dr Repsilber has no conflicts of interest to disclose. Dr Hyötyläinen has no conflicts of interest to disclose. Dr Høivik has served as a speaker and/or advisory board member for AbbVie, Ferring, Galapagos, MEDA, MSD, Pfizer, Takeda, and Tillotts Pharma. She has also received grant support from Ferring, Tillotts Pharma, Takeda, and Pfizer. Dr Halfvarson has served as speaker and/or advisory board member for AbbVie, Aqilion, BMS, Celgene, Celltrion, Dr Falk Pharma and the Falk Foundation, Ferring, Galapagos, Gilead, Hospira, Index Pharma, Janssen, MEDA, Medivir, MSD, Novartis, Pfizer, Prometheus Laboratories Inc., Sandoz, Shire, Takeda, Thermo Fisher Scientific, Tillotts Pharma, Vifor Pharma, UCB and received grant support from Janssen, MSD and Takeda.

## Additional information

[1]School of Medical Sciences, Faculty of Medicine and Health, Örebro University, Örebro, Sweden. [2]Department of Women's and Children's Health, Uppsala University, Uppsala, Sweden. [3]Department of Gastroenterology, Oslo University Hospital, Oslo, Norway and Faculty of Medicine, University of Oslo, Oslo, Norway. [4]Department of Clinical Research Laboratory, Faculty of Medicine and Health, Örebro University, Örebro, Sweden. [5]Department of Pediatrics and Adolescent Medicine, Akershus University Hospital, Lørenskog, Norway. [6]Department of Pediatrics, Vestfold Hospital Trust, Tønsberg, Norway. [7]Translational Gastroenterology Unit, Nuffield Department of Experimental Medicine, University of Oxford, Oxford, United Kingdom. [8]Biomedical Research Center, University of Oxford, Oxford, United Kingdom. [9]Department of Paediatric and Adolescence Medicine, Copenhagen University Hospital - Amager and Hvidovre, Hvidovre, Denmark. [10]Copenhagen Center for Inflammatory Bowel Disease in Children, Adolescents and Adults, Copenhagen University Hospital - Amager and Hvidovre, Hvidovre, Denmark. [11]Department of Pediatric Medicine, Oslo University Hospital, Oslo, Norway. [12]Department of Gastroenterology, Akershus University Hospital, Lørenskog, Norway and Faculty of Medicine, University of Oslo, Oslo, Norway. [13]Department of Gastroenterology, Telemark Hospital Trust, Skien, Norway. [14]Karolinska Institutet, Department of Medicine Solna, Stockholm, Sweden. [15]Karolinska University Hospital, Gastroenterology unit, Department of Gastroenterology, Dermatovenereology and Rheumatology, Stockholm, Sweden. [16]Department of Medical Sciences, Uppsala University, Uppsala, Sweden. [17]Department of Microbiology and Immunology, Institute of Biomedicine, Sahlgrenska Academy, University of Gothenburg, Gothenburg, Sweden. [18]Department of Biomedical and Clinical Sciences, Linköping University, Linköping, Sweden. [19]IKERBASQUE, Basque Foundation for Science, Bilbao, Spain. [20]Gastrointestinal Genetics Lab, CIC bioGUNE - BRTA, Derio, Spain. [21]Department of Medicine & Surgery, LUM University, Casamassima, Italy. [22]Turku Bioscience Centre, University of Turku and Åbo Akademi University, Turku, Finland. [23]Gastrounit, medical division, Copenhagen University Hospital - Amager and Hvidovre, Hvidovre, Denmark. [24]Department of Paediatrics, University of Oxford, Oxford, UK. [25]School of Science and Technology, Örebro University, Örebro, Sweden. [26]Department of Gastroenterology, Faculty of Medicine and Health, Örebro University, Örebro, Sweden. [27]These authors jointly supervised this work: Tuulia Hyötyläinen, Marte Lie Høivik, Jonas Halfvarson. ✉e-mail: jonas.halfvarson@regionorebrolan.se

