## [Peer Review File · Nature Communications]

Identification and validation of a blood-based diagnostic lipidomic signature of pediatric inflammatory bowel diseaseREVIEWER COMMENTS

Reviewer #1 (Remarks to the Author):

The authors present a study on identification, validation and comparison of a novel lipidomic signature for diagnosis of PIBD. The comparison to CRP and FcP, alongside validation in a distinct cohort is important. The study is of interest, has a clear clinical translation (although the emphasis on diagnosis vs screening might need to be considered) and is well written. I have comments and questions which may be addressed by the authors to clarify points.

Introduction-The authors present a good overview of the topic. I think the emphasis on diagnosis may be slightly confusing- is this a tool aiming to help with screening? Such as CRP or FcP? Diagnostic processes are through endoscopy, and thus this tool will not be replacing that method of diagnosis. CRP, for example, may be a useful as a (relatively) sensitive tool, but it is highly non-specific. FcP is a fantastic screening and monitoring tool, and stool tests are much more widely accepted, especially with postal kits and home testing kits available.

Methods- These are nicely described. I have some specific questions-

- Were cases and controls recruited at the same time points, most importantly blood samples achieved at the same point in the diagnostic pathway (rather than only cases recruited at a point after bowel prep, after fasting etc.)
- How long after the initial recruitment were non-IBD controls followed-up for?
- I am pleased the statistical methods are corrected for multiple testing
- Am i correct in thinking that largely all patients were pooled for analyses, regardless of age, phenotypic subtype, disease severity etc.?

Results-

- When models were employed what was the stability of the lipid profiles ascertained in the model? I.e. when cross-validation/random sampling was done were the same lipids always identified, or were different patterns identified each time?
- I think a NPV of 76%, whilst interesting, is likely difficult to apply in the clinic as a unique tool, however it could be used in the context of other tests. Further results indicate comparable performance to FcP in the AUC.
- For the controls, what conditions were diagnosed (if any), were all patients and controls screened for infection? Would these lipid signatures be replicated in other non-IBD inflammatory states (such as infection) in a similar way to FcP, or are they distinct?
- Was CRP only used in the analyses in the confirmed absence of GI or systemic infection?
- It is notable that both the discovery and validation cohorts' control children are younger than the IBD patients? How does metabolic maturation of lipids progress with age/puberty and could these models merely be detecting differences in metabolism related to age, rather than IBD vs controls?
- It would be useful to see a full comparison of the clinical, demographic features of IBD vs controls in table 1- including BMI etc. This has the potential to be a serious confounder.

Discussion-

The discussion is well written, focusing on the literature and the important strengths and limitations of the work. I think the authors would acknowledge the limitations set out above in any revision of the article.

- Whilst this is not the purpose of the study, what is the metabolic driver of these lipids, what could the pathogenesis be resulting in elevation/decrease? With FcP there is a clear process related to inflammation in the gut, confirming that it's a biologically relevant marker for disease. This is less clear here although is alluded to in the initial paragraph.
- Is there any work on these lipids (PC and LacCer) in systemic inflammation, or other inflammatory processes? Are these merely a marker of inflammation or are these specific to IBD?

Reviewer #2 (Remarks to the Author):

The authors have performed a comprehensive study of lipid omics in pediatric IBD. The patient sample characteristics shows inclusion of diverse patients. The data was analyzed by diverse mathematical and statistical perspective. Training and validation were performed according to acceptable standard. The data clearly shows the association of two newly discovered lipid species that compete well to clinically used hsCRP. The authors have reported a NPV of 76% compared to 40% for hsCRP which is significantly better. These results will create interest in the community on the value of these biomarkers in diagnosis of pediatric IBD.

Minor comments:

A Violin plots of the final model (combined) signature in SC vs IBD can be useful to include as part of a supplementary figure. The authors have included the violin plots for individual biomarker in Figure 4B.

The author should investigate new published studies PMID: 36662167 for discussion.

Reviewer #3 (Remarks to the Author):

In this work authors identified lipid features to diagnose pediatric IBD by analyzing plasma samples from a Swedish inception cohort of treatment-naïve pediatric patients with suspected IBD (n=94). Further they were validated in a Norwegian inception cohort (n=116) with serum as sample. They found two molecular lipid species, lactosyl ceramide (d18:1/16:0) and phosphatidylcholine (18:0p/22:6) had better diagnostic performance. This work is of meaningful.

Major comments,

1. It is unusual in the discovery set, plasma was used, but in the validation set serum was used. Although it is understandable, authors can't find the same samples. Because two lipids are used, the established equation and the cutting-off value will be influenced. Authors used a nontargeted method to analyze the samples, in the use of combination marker the equation has to be established, respectively in the discovery and validation stages.
2. The sample number was not big enough, especially in the UC group. It is not possible to have the reliable scientific conclusion. Is it possible to increase the sample number if authors want to define the reliable markers for subtyping of the IBD? This is why "For IBD vs symptomatic controls, three molecular lipid species could be replicated in the validation cohort. The corresponding numbers were two for CD and one for UC". The sample number should be greatly increased, otherwise the results are difficult to be repeated, and "The discrepancy between the previous findings and the results in this study may be explained by the use of metabolomics vs non-targeted lipidomics and differences in sample size" (lines 470-472).

Minor comments,

3. Internal standard mixture should be given.
4. In line 238, two "according to"
5. In fact because of difficult in the quality control, 32 molecular lipid species used to stratify the IBD from the control have no meaning.
6. Authors found that "the relationship of LacCer(d18:1/16:0), but not PC(18:1p/22:6), and IBD is influenced by age and BMI". In Table 1 the significance of age and BMI in different groups should be given. In the AUC calculations age and BMI should be adjusted.

Reviewer #4 (Remarks to the Author):

This paper proposed a blood-based diagnostic lipidomic signature for pediatric inflammatory bowel disease (IBD).

An important finding is that a diagnostic algorithm has been constructed with only two markers, lactosyl ceramide (d18:1/16:0) and phosphatidylcholine (18:0p/22:6).

As stated in Introduction, this study aims to develop biomarkers for early diagnosis, and evaluation based on AUC alone is considered inappropriate.

A marker that can be used to screen potential patients with a blood test quickly would be desirable.

Evaluating the specificity using a highly sensitive cutoff value (e.g., sensitivity = 90% or 95%) would be more appropriate.

It is difficult to determine whether the marker is suitable for early diagnosis based on analyzing the optimal cutoff by Youden's index.

This paper uses CRP alone as a comparison, but it would be better to compare it with other blood tests, Etc.

Reference 10 (Levine A et al., J Pediatr Gastroenterol Nutr 2014) cited in the Introduction deals with diagnostic methods recommended in children with suspected IBD.

This recommendation includes biomarkers (e.g., fecal calprotectin (FC) and lactoferrin) other than CRP.

"Recommendation. Initial blood tests should include a complete blood count, at least two inflammatory markers, albumin, transaminases, and gGT. Fecal calprotectin is superior to any blood marker for detecting intestinal inflammation (EL2, RGC)."

"Pediatric data exist primarily for FC and lactoferrin. Both markers are excellent tools for identifying the presence of intestinal inflammation with high sensitivity."

Both the test and validation cohorts in this study have small sample sizes.

Since sensitivity and specificity can be evaluated in a retrospective case-control study, it would be desirable to validate the results on a slightly larger scale of data.

It is important to note that this study includes a prospective cohort study.

In Table 2, a hypothesis test of the difference in AUC for additional biomarkers has been performed, but such an evaluation method is inappropriate.

It would be preferable to use and add the integrated discriminant improvement (IDI) to evaluate discrimination when a "new" model incorporates additional biomarkers and an "old" model without them.

- Pencina MJ, D'Agostino RB Sr, Demler OV. Novel metrics for evaluating improvement in discrimination: net reclassification and integrated discrimination improvement for normal variables and nested models. *Stat Med.* 2012;31(2):101-13. doi: 10.1002/sim.4348.

- Hayashi K, Eguchi S. The power-integrated discriminant improvement: An accurate measure of the incremental predictive value of additional biomarkers. *Stat Med.* 2019;38(14):2589-2604. doi:10.1002/sim.8135

**Reviewer #1 (Remarks to the Author):**

The authors present a study on identification, validation and comparison of a novel lipidomic
signature for diagnosis of PIBD. The comparison to CRP and FcP, alongside validation in a
distinct cohort is important. The study is of interest, has a clear clinical translation (although
the emphasis on diagnosis vs screening might need to be considered) and is well written. I
have comments and questions which may be addressed by the authors to clarify points.

Introduction-The authors present a good overview of the topic. I think the emphasis on
diagnosis may be slightly confusing- is this a tool aiming to help with screening? Such as
CRP or FcP? Diagnostic processes are through endoscopy, and thus this tool will not be
replacing that method of diagnosis. CRP, for example, may be a useful as a (relatively)
sensitive tool, but it is highly non-specific. FcP is a fantastic screening and monitoring tool,
and stool tests are much more widely accepted, especially with postal kits and home testing
kits available.

**Response:** We would like to thank the reviewer for the thoughtful comment. It is certainly
correct that the method is only to be used as a first screening tool in the diagnostic pathway
and not to replace endoscopy as the method of diagnosis. We have updated the introduction
and discussion to reflect this as indicated below:

Introduction

*...” Several plasma or serum biochemical markers have been investigated as biomarkers in*
*the diagnostic pathway in IBD, i.e., identifying those who should be referred for endoscopy*
*and further investigations.”...*

Discussion

*...” Taken together, our study suggests a role for LacCer(d18:1/16:0) and PC(18:0p/22:6) in*
*the pathophysiology of IBD and affirms the use of a blood-based lipidomic signature as a tool*
*to be used in combination with existing clinically established markers to rule out pediatric IBD*
*and guide referral for endoscopy and further investigations.”...*

Methods- These are nicely described. I have some specific questions-

- Were cases and controls recruited at the same time points, most importantly blood samples
achieved at the same point in the diagnostic pathway (rather than only cases recruited at a
point after bowel prep, after fasting etc.)

**Response:** We have revised the methods section clarifying that both IBD patients and
symptomatic controls in the discovery and validation cohort were included at the same point
in the diagnostic pathway i.e., before the endoscopic examination.

Methods

*...” Both patients with IBD and symptomatic controls were included at the same point in the*
*diagnostic pathway i.e., before the endoscopic examination.”...*

- How long after the initial recruitment were non-IBD controls followed-up for?

**Response:** None of the symptomatic controls in the discovery cohort were diagnosed with
IBD during a follow up period of ≥ 3 years.

- I am pleased the statistical methods are corrected for multiple testing

**Response:** We thank the reviewer for noting this.

- Am i correct in thinking that largely all patients were pooled for analyses, regardless of age,
phenotypic subtype, disease severity etc.?

**Response:** Yes, that is correct.

Results-

- When models were employed what was the stability of the lipid profiles ascertained in the
model? I.e. when cross-validation/random sampling was done were the same lipids always
identified, or were different patterns identified each time?

**Response:** The ML models were employed using $\alpha = 0.1$ and a lambda obtained by
optimization in internal 5-fold cross-validations. The selection of the molecular lipid signature
was based on 500 model fits, and all individual molecular lipid species with non-zero
coefficients in any model are reported. To illustrate this, we have added variable importance
scores to Figure 3.

Results

*... "Information about the variable importance projection (VIP) score for each molecular lipid
is provided in Figure 3." ...*

- I think a NPV of 76%, whilst interesting, is likely difficult to apply in the clinic as a unique
tool, however it could be used in the context of other tests. Further results indicate
comparable performance to FcP in the AUC.

**Response:** We agree that this signature could be used in combination with other tests. We
have highlighted this in the conclusion of the discussion.

Discussion

*... "Taken together, our study suggests a role for LacCer(d18:1/16:0) and PC(18:0p/22:6) in
the pathophysiology of IBD and affirms the use of a blood-based lipidomic signature as a tool
to be used in combination with existing clinically established markers to rule out pediatric IBD
and guide referral for endoscopy and further investigations." ...*

- For the controls, what conditions were diagnosed (if any), were all patients and controls
screened for infection?

**Response:** All individuals in each cohort followed the same diagnostic pathway and
underwent the same diagnostic investigations, including screening of infections. In the
cohorts, the symptomatic controls were diagnosed with various non-IBD conditions such as
celiac disease, infectious enteritis, food allergy, orofacial granulomatosis, and functional
gastrointestinal diseases, primarily diarrhea-prominent irritable bowel syndrome.

Would these lipid signatures be replicated in other non-IBD inflammatory states (such as
infection) in a similar way to FcP, or are they distinct?

**Response:** We would like to thank the reviewer for the insightful comment and agree that
this is a clinically relevant question. Theoretically, examination of pediatric patients with non-
IBD inflammatory states, including infections would be of great interest. Unfortunately, the
number of patients with infectious enteritis was too few to enable any meaningful
comparison. However, we performed a targeted analysis of LacCer(d18:1/16) and
PC(18:0p/22:6) in a third cohort of pediatric patients (n=263) from Norway, Denmark and UK.
In this cohort, 30 patients were diagnosed with celiac disease and 164 with IBD. Significant
differences in absolute concentrations of LacCer(d18:1/16) and PC(18:0p/22:6) were
observed when comparing celiac disease with IBD. We have added these novel data to the
revised method, results and discussion.

Method

See section "*Targeted confirmation of molecular lipid signature using UHPLC-MS/MS*".

Results

*... "To discern whether these molecular lipids serve as markers for inflammatory*
*gastrointestinal diseases generally or are more IBD specific, we compared patients with IBD*
*to the subset of celiac disease patients within the symptomatic controls. We observed*
*significantly increased concentrations of LacCer(d18:1/16:0) ($\beta = 1.29$, 95%CI 0.78,1.80,*
*$P < 0.001$) and numerically decreased concentrations of PC(18:1p/22:6) ($\beta = -0.42$, 95%CI -*
*0.86, 0.02, $P = 0.06$) in patients with IBD compared to patients with celiac disease."...*

Discussion

*... "In an independent third cohort, we confirmed the significant differences in the prioritized*
*molecular lipids (LacCer(d18:1/16:0) and PC(18:1p/22:6)) between patients with IBD and*
*symptomatic controls using a targeted absolute quantification method. Moreover, we*
*demonstrated that these molecular lipids were not broad markers of inflammation but*
*seemed to be more IBD specific."...*

- Was CRP only used in the analyses in the confirmed absence of GI or systemic infection?

**Response:** We apologize, gastrointestinal and other gastrointestinal diseases were
erroneously listed as exclusion criteria in the previous version of the manuscript. We have
omitted these diagnoses from exclusion criteria in the revised version of the manuscript. In
general, gastrointestinal infections were ruled out by the general practitioner before referral
to the pediatric departments. However, a few patients in the inception cohorts turned out to
be diagnosed with infectious enteritis during diagnostic workup by the pediatric
gastroenterologist. In the discovery cohort, CRP was assessed in clinical routine, whereas
high sensitivity CRP was assayed in a single batch for all patients in the validation cohort.

- It is notable that both the discovery and validation cohorts' control children are younger than
the IBD patients? How does metabolic maturation of lipids progress with age/puberty and
could these models merely be detecting differences in metabolism related to age, rather than
IBD vs controls?

**Response:** Thank you for highlighting this important aspect. We have augmented the
information about correlation between age and the two molecular lipids and inserted a figure
in the main manuscript showing this relationship (see Figure 3f). We have also highlighted
the analysis illustrating the moderating effect of age on the relationship between the two
molecular lipids and IBD (see Section Sensitivity analysis of short diagnostic signature
LacCer(d18:1/16:0) and PC(18:1p/22:6) and Figure 5a-b). Collectively, these data
demonstrate that the relationship of LacCer(d18:1/16:0), but not PC(18:1p/22:6), and IBD is
influenced by age. However, when adding age, sex, BMI, and albumin to the lipid signature,

no clinically significant improvement in diagnostic performance was observed (AUC 0.87 vs
0.89) as illustrated in Figure 4b.

- It would be useful to see a full comparison of the clinical, demographic features of IBD vs
controls in table 1- including BMI etc. This has the potential to be a serious confounder.

**Response:** We have included a comparison of clinical and demographic features between
IBD and symptomatic controls within both the Swedish and Norwegian cohorts in Table 1. As
outlined in our response to the comment above, figures illustrating the correlation and
interaction of BMI have been added to the main body of the manuscript (see Figure 3f and
Figure 5a-b).

Discussion-

The discussion is well written, focusing on the literature and the important strengths and
limitations of the work. I think the authors would acknowledge the limitations set out above in
any revision of the article.

**Response:** Thank you for the encouraging comment, we have acknowledged the
limitations above in the discussion.

Discussion

...”*To gain further mechanistic understanding, future studies should include patients
in remission and evaluate associations of disease activity and retrieve data from
follow-up visits of patients in these cohorts and examine the relationship of lipidomic
species with therapy response and long-term outcomes, preferably also integrating
additional omics data. For clinical translation of the molecular lipid signature, method
validation and including standard curve establishment using authentic and isotope-
labelled internal and injection standards as well as stability, repeatability,
reproducibility, and interlaboratory studies are required for clinical implementation as
well as regulatory approval. Furthermore, clinical cut-offs and corresponding
likelihood ratios for various clinical scenarios need to be established. Thus, further
work is required to ultimately translate our findings into an assay for clinical use.”...*

- Whilst this is not the purpose of the study, what is the metabolic driver of these
lipids, what could the pathogenesis be resulting in elevation/decrease? With FcP
there is a clear process related to inflammation in the gut, confirming that it's a
biologically relevant marker for disease. This is less clear here although is alluded to
in the initial paragraph.

**Response:** Thank you for your relevant comment, we have expanded the discussion
on their potential role in IBD pathogenesis,

Discussion

...” *The role of sphingolipids in the context of IBD is complex and the mechanisms
behind the increased levels of LacCer(d18:1/16:0) remain to be elucidated. Even
though we observed increased levels already at diagnosis, it is unclear whether this
finding precedes the transition from preclinical IBD to onset of symptoms and an IBD
diagnosis. Experimental studies have found various sphingolipids important for
plasma membrane stability and for signaling to several receptor molecules.²³ Lactosyl
ceramides have, for instance, been found to be significantly enriched in the apical
membrane of polarized intestinal epithelial cells.²⁴ Different pro-inflammatory factors
have been shown to activate lactosylceramide synthase to produce lactosyl*

*ceramides, which in turn activate mucosal cell differentiation and maturation.*²⁴
*Ceramides can also be transformed into ceramide 1-phosphate, or they can undergo*
*further degradation into sphingosine, which, in turn, can be phosphorylated to*
*produce sphingosine 1-phosphate (S1P). These molecules play a critical role in the*
*regulation of inflammatory processes, and recent drug developments have identified*
*S1P as a treatment target for IBD, modulating migration of lymphocytes from lymph*
*nodes.*²⁵ ...

...” *Ferru-Clément et al. recently identified several structurally unique lipids*
*(phosphatidylethanolamine ether (O-16:0/20:4), sphingomyelin (d18:1/21:0),*
*cholesterol ester (14:1), very long-chain dicarboxylic acid [28:1(OH)] and sitosterol*
*sulfate) with association to CD when compared to healthy controls, highlighting*
*multiple different biologic pathways including breakdown of intestinal homeostasis*
*and barrier integrity.*¹⁹ *Alkyl ether PCs, in addition to their structural roles in cell*
*membranes, are thought to function as endogenous antioxidants, and emerging*
*studies suggest that they are involved in cell differentiation and signaling pathways.*³³
*These lipids have shown to be endogenous antigens to activate invariant natural killer*
*T cells (iNKT),*³⁴ *and associated with autoimmunity.*³⁵ ...

-Is there any work on these lipids (PC and LacCer) in systemic inflammation, or other
inflammatory processes? Are these merely a marker of inflammation or are these specific to
IBD?

**Response:** As outlined above, we observed significant differences in absolute
concentrations of LacCer(d18:1/16) and PC(18:0p/22:6) were observed when comparing IBD
with celiac disease. These findings may potentially indicate that these lipids could in part be
specific to IBD and may not represent markers of inflammation alone. We have added these
results to the revised method, results and discussion.

**Reviewer #2 (Remarks to the Author):**

The authors have performed a comprehensive study of lipid omics in pediatric IBD. The patient
sample characteristics shows inclusion of diverse patients. The data was analyzed by diverse
mathematical and statistical perspective. Training and validation were performed according
to acceptable standard. The data clearly shows the association of two newly discovered lipid
species that compete well to clinically used hsCRP. The authors have reported a NPV of 76%
compared to 40% for hsCRP which is significantly better. These results will create interest in
the community on the value of these biomarkers in diagnosis of pediatric IBD.

**Response:** Thank you for the encouraging comment.

**Minor comments:**

A Violin plots of the final model (combined) signature in SC vs IBD can be useful to include
as part of a supplementary figure. The authors have included the violin plots for individual
biomarker in Figure 4B.

**Response:** The Reviewer has correctly pointed out that we have included violin plots to
illustrate the distribution of each molecular lipid (in the revised version Figure 4a). We would
be happy to further refine the plots. However, we struggle with the suggestion to include both
molecular lipids in a single violin plot. We would be happy to receive further guidance if the
Reviewer believes that it would be beneficial.

The author should investigate new published studies PMID: 36662167 for discussion.

**Response:** Thank you for bringing the study by Ferru-Clément et al. to our attention.
We have included it in the discussion section according to the reviewer's suggestion.

**Reviewer #3 (Remarks to the Author):**

In this work authors identified lipid features to diagnose pediatric IBD by analyzing plasma
samples from a Swedish inception cohort of treatment-naïve pediatric patients with
suspected IBD (n=94). Further they were validated in a Norwegian inception cohort (n=116)
with serum as sample. They found two molecular lipid species, lactosyl ceramide
(d18:1/16:0) and phosphatidylcholine (18:0p/22:6) had better diagnostic performance. This
word is of meaningful.

**Response:** Thank you for the encouraging feedback.

Major comments,

1. It is unusual in the discovery set, plasma was used, but in the validation set serum was
used. Although it is understandable, authors can't find the same samples. Because two
lipids are used, the established equation and the cutting-off value will be influenced. Authors
used a nontargeted method to analyze the samples, in the use of combination marker the
equation has to be established, respectively in the discovery and validation stages.

**Response:** We agree that molecular lipids in general may have different concentrations in
plasma vs serum. To address this issue, we have collected matched plasma and serum
samples and performed targeted analysis of the LacCer(d18:1/16:0) and PC(18:1p/22:6) and
can confirm that the concentration of these two analytes did not largely differ between the
two biological fluids. This information has been added to Supplementary information.

Supplemental Method

*"Distribution of LacCer(d18:1/16:0) and PC(18:0p/22:6) in serum vs plasma*
*Paired serum and plasma samples from healthy volunteers were analyzed to compare*
*LacCer(d18:1/16:0) and PC(18:0p/22:6) concentrations using the described methodology.*
*Results revealed no significant differences between serum and plasma concentrations for*
*both LacCer(d18:1/16:0) (369 ng/mL in serum vs 339 ng/mL in plasma, P = 0.13) and*
*PC(18:0p/22:6) (223 ng/mL in serum vs 218 ng/mL in plasma, P = 0.72). These findings*
*indicate no significant differences between paired plasma and serum concentrations."*

2. The sample number was not big enough, especially in the UC group. It is not possible to
have the reliable scientific conclusion. Is it possible to increase the sample number if authors
want to define the reliable markers for subtyping of the IBD? This is why "For IBD vs
symptomatic controls, three molecular lipid species could be replicated in the validation
cohort. The corresponding numbers were two for CD and one for UC". The sample number
should be greatly increased, otherwise the results are difficult to be repeated, and "The
discrepancy between the previous findings and the results in this study may be explained by
the use of metabolomics vs non-targeted lipidomics and differences in sample size" (lines
470-472).

**Response:** We agree that a larger sample size, i.e., patient population, would increase the
possibility to identify molecular lipids that are associated with IBD and especially UC.
However, signatures of many lipids may preclude their translation to clinical practice. To
strengthen our findings from the discovery and validation, we have now performed a targeted
analysis of absolute concentrations of LacCer(d18:1/16) and PC(18:0p/22:6) in a third cohort
of pediatric patients (n=263) from Norway, Denmark, and UK. We could demonstrate that the
comparison of IBD with symptomatic controls was consistent with our previous results. Also,
as correctly pointed out by the Reviewer, separate analysis of CD and UC showed significant
differences compared to symptomatic controls when examining this large third cohort, except
for PC(18:0p/22:6), in the comparison of UC vs symptomatic controls (see Figure 6). These
results provide additional confirmation that LacCer(d18:1/16) and PC(18:0p/22:6)
concentrations serve as reliable markers of IBD. We have added these new results to the
revised method, results, and discussion sections.

As noted by the Reviewer, differences in metabolomics platforms and sample size
probably explains why we could identify several differentially associated lipids in our analysis
of the Swedish regional cohort (see Figure 2a), whereas this was not the case when
analysing plasma from a subset of patients in the ancillary study (Nyström et al., 2022).

Minor comments,

3. Internal standard mixture should be given.

**Response:** We have revised according to the Reviewer's suggestion.

Methods

...*The internal standard solution contained the following compounds:*
*phosphatidylethanolamine (PE(17:0/17:0)), sphingomyelin (SM(d18:1/17:0)), ceramide*
*(Cer(d18:1/17:0)), phosphatidylcholine (PC(17:0/17:0)), lysophosphatidylcholine (LPC(17:0))*
*and lysophosphatidylcholine (PC(16:0/d31/18:1)), were purchased from Avanti Polar Lipids,*
*Inc. (Alabaster, AL, USA) and triheptadecanoylglycerol (TG(17:0/17:0/17:0)), and cholesteryl*
*ester (CE17:0) were purchased from Larodan AB (Solna, Sweden). The calibration curve*
*solutions contained the following compounds: LPC (18:0), cholesteryl ester (18:1, 9Z),*
*Cer(d18:1/24:0), Cer(d18:0/18:1, 9Z), triglyceride (16:0/16:0/16:0), PC(16:0/16:0),*
*Triglyceride (18:0/18:0/18:0), CE(18:0), LPC(18:1), LPE(18:1), PC(16:0/18:1),*
*Cer(d18:1/18:1, 9Z), PC(18:0/18:0), PE(16:0/18:1), CE(18:2, 9Z, 12Z), CE(16:0),*
*DG(18:1)."*...

4. In line 238, two "according to"

**Response:** We have deleted this duplicate.

5. In fact because of difficult in the quality control, 32 molecular lipid species used to stratify
the IBD from the control have no meaning.

**Response:** We agree that the 32 molecular lipid species are of limited value for clinical
translation but have kept this information since these lipids may disclose novel biological
mechanisms related to development of IBD. Moreover, we have expanded the discussion on
potential disease mechanisms related to our findings.

Discussion

...” *The fact that two of these cohorts were represented by only treatment-naïve children*
*demonstrates that the increase occurs already at diagnosis. We further extended these*
*findings by showing that the association of LacCer(d18:1/16:0) with IBD was most*
*pronounced in older pediatric patients and in those with a higher BMI. These interactions*
*have not been reported previously and are likely attributed to biological factors linked to*
*childhood growth, development, and changing physiology. The role of sphingolipids in the*
*context of IBD is complex and the mechanisms behind the increased levels of*
*LacCer(d18:1/16:0) remain to be elucidated. Even though we observed increased levels*
*already at diagnosis, it is unclear whether this finding precedes the transition from preclinical*
*IBD to onset of symptoms and an IBD diagnosis. Experimental studies have found various*
*sphingolipids important for plasma membrane stability and for signaling to several receptor*
*molecules.²³ Lactosyl ceramides have, for instance, been found to be significantly enriched in*
*the apical membrane of polarized intestinal epithelial cells.²⁴ Different pro-inflammatory*
*factors have been shown to activate lactosylceramide synthase to produce lactosyl*
*ceramides, which in turn activate mucosal cell differentiation and maturation.²⁴ Ceramides*
*can also be transformed into ceramide 1-phosphate, or they can undergo further degradation*
*into sphingosine, which, in turn, can be phosphorylated to produce sphingosine 1-phosphate*
*(S1P). These molecules play a critical role in the regulation of inflammatory processes, and*
*recent drug developments have identified S1P as a treatment target for IBD, modulating*
*migration of lymphocytes from lymph nodes.²⁵ ...*

...” *Ferru-Clément et al. recently identified several structurally unique lipids*
*(phosphatidylethanolamine ether (O-16:0/20:4), sphingomyelin (d18:1/21:0), cholesterol*
*ester (14:1), very long-chain dicarboxylic acid [28:1(OH)] and sitosterol sulfate) with*
*association to CD when compared to healthy controls, highlighting multiple different biologic*
*pathways including breakdown of intestinal homeostasis and barrier integrity.¹⁹ Alkyl ether*
*PCs, in addition to their structural roles in cell membranes, are thought to function as*
*endogenous antioxidants, and emerging studies suggest that they are involved in cell*
*differentiation and signaling pathways.³³ These lipids have shown to be endogenous antigens*
*to activate invariant natural killer T cells (iNKT),³⁴ and associated with autoimmunity.³⁵*
*Collectively, our findings of depletion of plasma and serum PC(18:0p/22:6) in pediatric IBD*
*may act as a potential treatment target. This hypothesis is supported by the finding that PC-*
*rich phospholipid supplementation (6g daily) over three months resulted in an overall*
*decreased inflammatory activity in patients with UC.³⁶ ...*

6. Authors found that “the relationship of LacCer(d18:1/16:0), but not PC(18:1p/22:6), and
IBD is influenced by age and BMI”. In Table 1 the significance of age and BMI in different
groups should be given. In the AUC calculations age and BMI should be adjusted.

**Response:** Table 1 has been revised as suggested. Also, we have added age, BMI, sex,
and albumin (as suggested by Reviewer 4) to the signature (see Figure 5b). As illustrated, no
clinically significant improvement in diagnostic performance was observed when adding
these covariates (AUC 0.87 vs 0.89).

**Reviewer #4 (Remarks to the Author):**

This paper proposed a blood-based diagnostic lipidomic signature for pediatric inflammatory
bowel disease (IBD). An important finding is that a diagnostic algorithm has been constructed
with only two markers, lactosyl ceramide (d18:1/16:0) and phosphatidylcholine (18:0p/22:6).

As stated in Introduction, this study aims to develop biomarkers for early diagnosis, and evaluation based on AUC alone is considered inappropriate. A marker that can be used to screen potential patients with a blood test quickly would be desirable. Evaluating the specificity using a highly sensitive cutoff value (e.g., sensitivity = 90% or 95%) would be more appropriate. It is difficult to determine whether the marker is suitable for early diagnosis based on analyzing the optimal cutoff by Youden's index.

Response: We would like to thank the reviewer for the thoughtful comment. In line with our response to Reviewer 1, we would like to highlight that the molecular lipid signature is only to be used as a first screening tool in the diagnostic pathway and not supposed to replace endoscopy as the method of diagnosis. However, the signature is not supposed to be applied as a screening instrument for the general population. Moreover, we agree that a highly sensitive blood-test would be desirable for screening as long as its specificity remains reasonable. But the tradeoff between sensitivity and specificity may differ between various clinical scenarios (such as screening for disease in the general population, identifying patients at primary care level who should be referred for further investigations, or identifying patients at the secondary care level). In our revised introduction and updated discussion, we have clarified that the molecular lipid signature has the potential to complement existing markers when assessing patients presenting gastrointestinal symptoms suggestive of possible IBD. We have highlighted this in the conclusion.

Conclusion

...”Taken together, our study suggests a role for LacCer(d18:1/16:0) and PC(18:0p/22:6) in the pathophysiology of IBD and affirms the use of a blood-based lipidomic signature as a tool to be used in combination with existing clinically established markers to rule out pediatric IBD and guide referral for endoscopy and further investigations.”...

This paper uses CRP alone as a comparison, but it would be better to compare it with other blood tests, Etc. Reference 10 (Levine A et al., J Pediatr Gastroenterol Nutr 2014) cited in the Introduction deals with diagnostic methods recommended in children with suspected IBD. This recommendation includes biomarkers (e.g., fecal calprotectin (FC) and lactoferrin) other than CRP. "Recommendation. Initial blood tests should include a complete blood count, at least two inflammatory markers, albumin, transaminases, and gGT. Fecal calprotectin is superior to any blood marker for detecting intestinal inflammation (EL2, RGC)." "Pediatric data exist primarily for FC and lactoferrin. Both markers are excellent tools for identifying the presence of intestinal inflammation with high sensitivity."

Response: We agree that guidelines on the diagnosis of IBD include various laboratory tests. However, several of these tests (transaminases and gGT) are used to identify patients with disease complications. Among above mentioned blood tests, albumin probably has the greatest capacity to differentiate IBD from symptomatic controls i.e., patients with other diagnoses. Therefore, we have measured albumin and added these results to the revised manuscript (see Figure 5b).

Both the test and validation cohorts in this study have small sample sizes. Since sensitivity and specificity can be evaluated in a retrospective case-control study, it would be desirable to validate the results on a slightly larger scale of data. It is important to note that this study includes a prospective cohort study.

**Response:** We agree that the number of patients in the discovery and validation cohorts
may seem low, but pediatric IBD is an uncommon condition. To address the limitations with
sample size, we have now performed a targeted analysis of absolute concentration
LacCer(d18:1/16) and PC(18:0p/22:6) in a third larger cohort of pediatric patients (n=263)
from Norway, Denmark, and UK. These new results further confirm that concentrations of
LacCer(d18:1/16) and PC(18:0p/22:6) serve as classifiers of pediatric IBD vs symptomatic
controls. The consistent observations across three distinct cohorts; discovery (Sweden,
n=94), validation (Norway, n=117), and confirmation (Norway, Denmark, and UK, n=263),
enhances the robustness of our findings. We have now added these new results in the
revised method, results, and discussion sections.

Even though the analysis of the third cohort confirmed our findings, establishment of cut-offs
for clinical use requires further assay development. We have elaborated on these aspects of
clinical translation in the discussion.

Discussion

*...”For clinical translation of the molecular lipid signature, method validation and including*
*standard curve establishment using authentic and isotope-labelled internal and injection*
*standards as well as stability, repeatability, reproducibility, and interlaboratory studies are*
*required for clinical implementation as well as regulatory approval. Furthermore, clinical cut-*
*offs and corresponding likelihood ratios for various clinical scenarios need to be established.*
*Thus, further work is required to ultimately translate our findings into an assay for clinical*
*use.”...*

In Table 2, a hypothesis test of the difference in AUC for additional biomarkers has been
performed, but such an evaluation method is inappropriate.
It would be preferable to use and add the integrated discriminant improvement (IDI) to
evaluate discrimination when a "new" model incorporates additional biomarkers and an "old"
model without them.

- Pencina MJ, D'Agostino RB Sr, Demler OV. Novel metrics for evaluating improvement in
discrimination: net reclassification and integrated discrimination improvement for normal
variables and nested models. *Stat Med.* 2012;31(2):101-13. doi: 10.1002/sim.4348.

- Hayashi K, Eguchi S. The power-integrated discriminant improvement: An accurate
measure of the incremental predictive value of additional biomarkers. *Stat Med.*
2019;38(14):2589-2604. doi:10.1002/sim.8135

**Response:** We agree that the employing these methods provides additional important
information about the capacity of the two molecular lipids for discriminating between patients
with IBD and symptomatic controls in comparison to hsCRP alone.

We have assessed the net reclassification index (NRI) and the integrated discriminant
improvement (IDI) for a model based on hsCRP with the addition of the two molecular lipids
and present the findings in the results, including Table 4, and in the discussion.

Methods

*...”Reclassification was assessed using net reclassification index (NRI) and integrated*
*discrimination improvement (IDI).⁴⁹...*

Results

...” To further assess the clinical relevance of the short lipidomic signature, we also evaluated
its capacity to reclassify patients with IBD vs symptomatic controls in the validation cohort.

*The addition of LacCer(d18:1/16:0) and PC(18:0p/22:6) to hsCRP, significantly improved*
*reclassification, as demonstrated by analysis of both NRI and IDI (P <0.001 for both) (Table*
*4). Evaluating the net reclassification impact of LacCer(d18:1/16:0) and PC(18:0p/22:6), we*
*observed a substantial improvement of 11% in reclassification of cases with IBD and 14% in*
*reclassification of symptomatic controls, reflecting their dual contribution. This indicates an*
*improved clinical utility of the molecular lipid signature over hsCRP alone.”...*

Discussion

...” For clinical translation, we demonstrated that a signature of only two molecular lipid
species, i.e., LacCer(d18:1/16:0) and PC(18:0p/22:6), was superior to hsCRP and the
addition of these molecular lipids to hsCRP, significantly improved the reclassification of
patients with IBD from symptomatic controls in the validation cohort.”...

REVIEWER COMMENTS

Reviewer #1 (Remarks to the Author):

The authors have extensively revised their original manuscript in line with the author comments. I am satisfied that most of these points have been addressed.

I have one outstanding question related to the ability of the signature to differentiate between cases/controls. Specifically, is this a signature related to gut inflammation per se, or specifically to IBD. The role of calprotectin here may be useful- does the lipid profile correlate to calprotectin in the non-IBD cases (such as infectious cases).

There is a hint this might be the cases, with younger children having a more pronounced predictive effect of the lipid signature, and a known association with high calprotectin values in younger children (reflecting normal intestinal immune maturation).

I think this is a key point- are we talking about a biomarker of intestinal inflammation (like calprotectin), or something specific to IBD. Without clarification of these data I think this limitation must be mentioned in the abstract and discussion

Reviewer #2 (Remarks to the Author):

The authors have provided adequate response to all the reviewers.

Reviewer #3 (Remarks to the Author):

Authors have addressed all of my questions. I have no other comments.

Reviewer #4 (Remarks to the Author):

The authors responded appropriately to my comments.
I appreciate the authors' efforts.

I have one minor comment about my (reviewer #4) first comment.

While I understand the authors' opinion, interpreting sensitivity and specificity under a cutoff value based on Youden's index is rarely valuable for clinical practice.

Table 3 should be modified to show the sensitivity or specificity for hsCRP and PC(18:0p/22:6)+LacCer(d18:1/16:0) for high sensitivity or high specificity cases (e.g., Se = 90%, 95% and Sp = 90%, 95%), respectively.

**Reviewer #1 (Remarks to the Author):**

The authors have extensively revised their original manuscript in line with the author
comments. I am satisfied that most of these points have been addressed.

I have one outstanding question related to the ability of the signature to differentiate between
cases/controls. Specifically, is this a signature related to gut inflammation per se, or
specifically to IBD. The role of calprotectin here may be useful- does the lipid profile correlate
to calprotectin in the non-IBD cases (such as infectious cases).

There is a hint this might be the cases, with younger children having a more pronounced
predictive effect of the lipid signature, and a known association with high calprotectin values
in younger children (reflecting normal intestinal immune maturation).

I think this is a key point- are we talking about a biomarker of intestinal inflammation (like
calprotectin), or something specific to IBD. Without clarification of these data, I think this
limitation must be mentioned in the abstract and discussion.

**Response:** We would like to thank the reviewer for carefully reviewing our revised
manuscript. We have performed the proposed analysis and added the results to the revised
manuscript.

**Results**

*... "In order to examine whether the molecular lipids reflect neutrophil activity and gut
inflammation per se or are specific to IBD, we assessed the correlation between the
molecular lipids and fecal calprotectin levels in the symptomatic controls only. However, no
statistically significant correlations were observed between fecal calprotectin and
LacCer(d18:1/16:0) ($r = 0.28$, $P = 0.13$), or PC(18:1p/22:6) ($r = 0.21$, $P = 0.25$)."*

**Discussion**

*... "Although we were unable to clearly demonstrate that the molecular lipid signature is
unique to IBD, the finding of different concentrations of LacCer(d18:1/16:0) and
PC(18:1p/22:6) in patients with IBD compared to patients with celiac disease (another
inflammatory disease) indicates that these are not general markers of inflammation. These
findings were further supported by the absence of significant correlations between the two
molecular lipids and fecal calprotectin levels among symptomatic controls only."...*

**Reviewer #4 (Remarks to the Author):**

The authors responded appropriately to my comments.
I appreciate the authors' efforts.

I have one minor comment about my (reviewer #4) first comment.

While I understand the authors' opinion, interpreting sensitivity and specificity under a cutoff
value based on Youden's index is rarely valuable for clinical practice.

Table 3 should be modified to show the sensitivity or specificity for hsCRP and
PC(18:0p/22:6)+LacCer(d18:1/16:0) for high sensitivity or high specificity cases (e.g., Se =
90%, 95% and Sp = 90%, 95%), respectively.

**Response:** We appreciate the Reviewer's thorough evaluation of our revised manuscript. As
recommended by the reviewer we have added information about the performance of the
signature and of hsCRP at 90% sensitivity and specificity in Table 3 and highlighted these
results in the text. Regarding applying a fixed sensitivity and specificity of 95%, the discrete
nature of the ROC curve based on 117 samples in the cohort did not allow an accurate
comparison for these levels. For the 90% level we chose to report the least favorable
statistics for the molecular lipid signature as a conservative approach.

**“Table 3. Youden index, sensitivity, specificity, positive likelihood ratio (LR), and**
**negative LR of hsCRP compared with two molecular lipids, LacCer(d18:1/16:0) and**
**PC(18:0p/22:6), in predicting pediatric inflammatory bowel disease in the validation**
**cohort. The first two rows represent the diagnostic test statistics based on optimal**
**Youden index. Rows three to six show statistics based on fixed sensitivity at 90% and**
**a fixed specificity at 90%.**

Evaluated model	Youden index (J)	Sensitivity (%)	Specificity (%)	LR(+)	LR(-)
hsCRP	0.42	67.5	70.3	2.3	0.5
PC(18:0p/22:6) and LacCer(d18:1/16:0)	0.66	83.8	78.4	3.9	0.2
hsCRP	NA	90.0	35.1	1.4	0.3
PC(18:0p/22:6) and LacCer(d18:1/16:0)	NA	90.0	67.6	2.8	0.1
hsCRP	NA	29.4	90.0	3.2	0.8
PC(18:0p/22:6) and LacCer(d18:1/16:0)	NA	66.3	90.0	7.1	0.4

71
72 Abbreviations: hsCRP, high sensitivity C-reactive protein, LR(+), likelihood ratio for positive
73 test result; LR(-), likelihood ratio for a negative test result”

74

REVIEWERS' COMMENTS

Reviewer #1 (Remarks to the Author):

All points addressed, recommend to accept

Reviewer #4 (Remarks to the Author):

I have no additional comments. Thank you.